# Private and Personalized Frequency Estimation in a Federated Setting

**Amrith Setlur**[*]
Carnegie Mellon University
asetlur@cs.cmu.edu

**Vitaly Feldman**
Apple
vitaly.edu@gmail.com

**Kunal Talwar**
Apple
ktalwar@apple.com

## Abstract

Motivated by the problem of next word prediction on user devices we introduce and study the problem of *personalized frequency histogram* estimation in a federated setting. In this problem, over some domain, each user observes a number of samples from a distribution which is specific to that user. The goal is to compute for all users a personalized estimate of the user's distribution with error measured in KL divergence. We focus on addressing two central challenges: *statistical heterogeneity* and *protection of user privacy*. Our approach to the problem relies on discovering and exploiting similar subpopulations of users which are often present and latent in real-world data, while minimizing user privacy leakage at the same time. We first present a non-private clustering-based algorithm for the problem, and give a provably joint differentially private version of it with a private data-dependent initialization scheme. Next, we propose a simple data model which is based on a mixture of Dirichlet distributions, to formally motivate our non-private algorithm and demonstrate some properties of its components. Finally, we provide an extensive empirical evaluation of our private and non-private algorithms under varying levels of statistical and size heterogeneity on the Reddit, StackOverflow, and Amazon Reviews datasets. Our results demonstrate significant improvements over standard and clustering-based baselines, and in particular, they show that it is possible to improve over direct personalization of a single global model.

## 1 Introduction

Federated learning algorithms jointly learn from decentralized user data, addressing statistical challenges in local learning [49, 58]. At the same time it presents two key challenges, among few others [44, 52]. Firstly, data distributions can vary across users (often called statistical heterogeneity [27, 76]), which reduces the effectiveness of a single, shared model for all users. Secondly, the data may be sensitive and mutually beneficial collaboration may compromise user privacy [31, 71]. An important practical problem where user data distributions are diverse, and user-privacy is paramount is next-word prediction for the keyboard input on user devices. Users often have diverse vocabulary, writing styles, and topics that leads to varied data distributions [32, 36, 65]. Naturally, frequency estimation is also one of the most basic statistical tasks with numerous other applications [14, 47].

Motivated by personalized next-word prediction, **we introduce and study the problem of personalized frequency estimation in a federated setting**. In this problem, each user has very few $(O(d))$ samples from an unknown, user-specific distribution over a large, finite domain of size $d$. The users interact with a server that needs to provide the user with a personalized estimate of their distribution. We focus on error measured in KL divergence that is common in language modeling [3, 38], and is equivalent to minimizing the negative log-likelihood of samples from the user distribution. For more general loss minimization problems, numerous personalized federated learning

---

[*]Part of the work was done during an internship at Apple.

38th Conference on Neural Information Processing Systems (NeurIPS 2024).

(PFL) algorithms [67, 76], tackle statistical heterogeneity by first learning a single global model across all users (for *e.g.*, FedAvg [57]), and then finetuning (FT) the global model locally for each user (FedAvg+FT [18, 20]), thus balancing collaboration and personalization. The downside of this approach is that it is agnostic to the structure of the user population which is highly heterogeneous, but often consists of a number of concentrated subpopulations of similar users.

**Statistical heterogeneity:** We study the personalized frequency estimation problem under an intuitive model of the user-population which consists of well-concentrated clusters of users where users in the same cluster have "similar" token distributions. For this, we propose an iterative algorithm that adapts Lloyd's clustering [53] to distribution estimation in KL divergence. Relying on the optimal competitive estimation rates for *local* Good-Turing estimators, we propose to estimate *global* cluster centers by averaging user-level Good-Turing estimates for the current users in the cluster. Furthermore, since the performance of clustering is significantly affected by the initialization [5] of cluster centers, we also propose a data-dependent initialization approach, also specific to estimation in KL. We improve the practicality of our algorithm by separately handling data-poor users in cases where users vary significantly in their local data sizes (size-heterogeneity).

**Privacy:** To address the challenge of user-level privacy we give a joint differentially private (DP) version of our iterative algorithm which requires that the estimator for user $i$ is differentially private with respect to the data of all the other users (but may depend arbitrarility on user $i$'s data) [48]. This definition is necessary due to the final personalization step which happens locally on user device and thus does not present a privacy risk. Specifically, we make each iteration of our clustering algorithm provably private by relying on adaptive clipping, secure aggregation, and common noise addition mechanisms [23]. Initialization of clustering algorithms typically uses data points themselves as initial centers and hence presents a significant challenge for privacy preserving analysis. Our provably private data-dependent initialization algorithm for cluster centers runs the exponential mechanism over candidates randomly sampled around the estimated population mean.

**Empirical evaluation:** We validate both non-private and private versions of our algorithm on the real world data datasets: Reddit [16], StackOverflow [6], and Amazon Reviews [62], where Reddit has a token vocabulary of size 10k and the others have 32k tokens. We find that our method furnishes significant gains over standard and clustering baselines, reducing the error by over $26\%$ and $42\%$ (averaged over datasets) in the non-private and private settings respectively. In the private, size-heterogeneous setting, we improve over the clustering baseline IFCA [33] by over $30\%$. Finally, we show that our data model inspired adjustments are pivotal in yielding the performance improvements in practice. Particularly, we justify the use of our Good-turing based estimator, and our private initialization scheme with a favorable privacy/utility tradeoff.

**Formal guarantees.** While our focus is on the empirical performance, to guide intuition and design of algorithms, we introduce a relatively simple generative model where user distributions are sampled from a mixture of Dirichlets, where Dirichlets are sufficiently separated globally and concentrated locally. We derive and analyze different estimators (*e.g.*, Bayes optimal, FedAvg [57], Good-Turing [34]). Even when the cluster identities are known, we show that FedAvg (average of user empirical distributions) has poor guarantees compared to our proposed estimator (average of user Good-Turing estimates), where the minimax error rates for the latter do not scale with dimension, and is optimal for some regimes of problem parameters. Thus, using this model we demonstrate how additional structure in the user data can be exploited in the context of our more concrete frequency estimation problem while preserving the privacy of user data.

## 2   Problem definition

We use $P$ to denote a distribution, $P[v]$ for the probability of event $v$, $\tilde{P}$ for an estimate of $P$ (but $\hat{P}$ if the estimate is computed without user collaboration), $\mathcal{P}$ for sets. $\mathbf{P}$ for matrices ($\mathbf{P}_{:,i}/\mathbf{P}_{i,:}$ index into the $i^{\text{th}}$ column/row respectively), and $\{P_i\}$ for a collection of elements $P_i$ indexed by $i$.

**Setup.** Let $\mathcal{V}$ be a finite vocabulary (set) of tokens, with vocabulary size $d =: |\mathcal{V}|$, and $\Delta(\mathcal{V})$ is the set of all probability distributions over $\mathcal{V}$. In a federated setup we have a collection of users $\mathcal{U}$, and each user $u \in \mathcal{U}$ has an unknown distribution $Q_u \in \Delta(\mathcal{V})$ over the tokens. The data available to each user $u$ is a dataset $\mathcal{S}_u$ comprising of $m_u$ i.i.d. samples from $Q_u$. We will use $\widehat{Q}_u$ to refer to the empirical distribution estimated from $\mathcal{S}_u$. The combined collection of datasets is referred to as meta-dataset $\mathcal{S} =: \{\mathcal{S}_u\}$, and similarly $\hat{\mathcal{Q}} =: \{\widehat{Q}_u\}_u$ is the collection of all empirical distributions.

Unless specified otherwise, we assume that we are in the *size-homogeneous* setting where all users have a dataset of the same size $m = m_u$. We refer to the more general case as *size-heterogeneous*.

**Goal.** Given the meta-dataset $\mathcal{S}$, the goal is to compute estimates $\{\widetilde{Q}_u\}_u$ such that:

1. *utility*: $\sum_u \mathbb{E}[\text{KL}(Q_u \| \widetilde{Q}_u)]$ is minimized, where the expectation is taken over sampling of meta-dataset $\mathcal{S}$ and any randomness in computing $\{\widetilde{Q}_u\}_u$.

2. *privacy*: $\widetilde{Q}_u$ is computed privately with respect to all users except $u$, *i.e.* the estimate is *joint differentially private* (see Definition 2.2).

**Definition 2.1** ($u$-neighboring meta-datasets)**.** Meta-datasets $\mathcal{S}$ and $\mathcal{S}'$ are $u$-neighbors, *i.e.*, $\mathcal{S} \sim_u \mathcal{S}'$ if they differ only in inclusion or deletion of user $u$'s private data. For any algorithm $\mathfrak{M}$, we denote $\mathfrak{M}_{-u}$ as the output of the algorithm to all other users except $u$.

**Definition 2.2** (Joint Differential Privacy [48])**.** Given $\varepsilon \geq 0$, $\delta \in (0,1]$, and neighbouring relation $\sim_u$, a randomized mechanism $\mathfrak{M} : \mathcal{D} \rightarrow \mathcal{Y}$ from the set of meta-datasets $\mathcal{D}$ to an output space $\mathcal{Y}$ is $(\epsilon, \delta)$-joint differentially private if for all $u$-neighboring $\mathcal{S} \sim_u \mathcal{S}'$, and all events $E \subseteq \mathcal{Y}$,

$$\Pr[\mathfrak{M}_{-u}(\mathcal{S}) \in E] \leq e^\epsilon \cdot \Pr[\mathfrak{M}_{-u}(\mathcal{S}') \in E] + \delta, \tag{1}$$

where probabilities are taken over the randomness of $\mathfrak{M}$. We say that $\mathfrak{M}$ is $\rho-zero\text{-}concentrated$ *joint DP* ($\rho$-zCJDP) if for all $u$-neighboring meta-datasets $\mathcal{S} \sim_u \mathcal{S}'$:

$$D_\alpha(\mathfrak{M}_{-u}(\mathcal{S}) \| \mathfrak{M}_{-u}(\mathcal{S}')) \leq \rho\alpha, \tag{2}$$

where $D_\alpha(\mathfrak{M}_{-u}(\mathcal{S}) \| \mathfrak{M}_{-u}(\mathcal{S}'))$ is the $\alpha$-Rényi Divergence between the outputs of all users except $u$.

## 3   Private algorithm for learning personalized histograms

In this section, we present our algorithm for learning private and personalized histograms in both size-homogeneous and size-heterogeneous settings. Without assumptions on the distribution of users, in the worst case, the best estimator would simply estimate each user's histogram locally, *i.e.*, estimate each user's $Q_u$ using their local dataset $\mathcal{S}_u$. Avoiding such worst-cases, we assume that the real-world problem is more structured. Specifically, we assume that each user distribution $Q_u$ is well clustered around $K$ unknown cluster centers in the population of users. Note, we assume no knowledge of the cluster centers or number of clusters $K$. In Section 5, we empirically validate the presence of clusters in real-world datasets by evaluating the performance of our clustering based algorithm on the same datasets, and in Section 4 we theoretically analyze some of the algorithmic design choices we make, in a stylized Bayesian setting that simulates a user-distribution with clusters.

As a warmup, we first discuss our local finetuning algorithm, where the setup is simpler. Say an algorithm runs a private collaborative protocol, and hands a distribution $P$ to user $u$, with the guarantee that $\text{KL}(Q_u \| P)$ is "reasonably" small, then how should the user $u$ finetune $P$ using the local dataset $\mathcal{S}_u$ available to them? For this question, we now present our local finetuning algorithm.

### 3.1   User-level local finetuning

Let $P$ be the distribution over tokens returned by a collaborative algorithm that uses datasets across all users to estimate a single distribution for either all users in the population, or all users in a specific cluster. In the next section, we discuss how to estimate $P$. Here, we discuss how to personalize $P$ for each user. For a user $u$, with empirical distribution $\widehat{Q}_u$ from $m_u$ samples, we use $\text{FT}(P, \widehat{Q}_u)$ to denote the personalized, locally finetuned estimate of user distribution $Q_u$ (see Eq. 3). Note that $P$ is moved closer to $\widehat{Q}_u$ in KL divergence (depending on choice of $\alpha > 0$). When the population of users is well concentrated, FT lowers variance by relying more on the global estimate $P$, while adding some bias ($\widehat{Q}_u$ is an unbiased estimate of $Q_u$, but $P$ is biased). In Section 4, we show that FT is the Bayes optimal estimator in KL divergence, under certain assumptions on the user distribution.

$$\text{FT}(P, \widehat{Q}_u) =: \frac{\alpha}{\alpha + m_u} \cdot P + \frac{m_u}{\alpha + m_u} \cdot \widehat{Q}_u \qquad \text{(local finetuning)} \tag{3}$$

Next, we discuss how to collaboratively estimate $P$. For a user-distribution with multiple clusters, a natural approach is to first identify the groups of clustered users. Then, we can use the distribution that is close in KL distance, and in expectation over the distribution of users in the cluser. We refer to these as the cluster centers. In the next Section, we present our algorithm to learn cluster centers, which we personalize for each user in the cluster using our local finetuning update (Equation 3).

## 3.2 Learning clusters in the population of size-homogeneous users

As mentioned in the start of Section 3, we base our approach on the belief that the user-population comprises of unknown clusters. Thus, given the collection of size-homogeneous users $\mathcal{U}$, the goal is to learn a user partition $\{\tilde{z}_u\}_u$, and the matrix of $K$ cluster centers $\tilde{\mathbf{P}} \in \mathbb{R}^{d \times K}$. To learn cluster centers $\tilde{\mathbf{P}}$, akin to the popular clustering algorithms like Lloyds [53], the objective we optimize is:

$$\min_{\{\tilde{P}_c\}_{c \in [K]}} \sum_{u \in \mathcal{U}} \min_{c \in [K]} \mathrm{KL}(Q_u \| \tilde{P}_c), \tag{4}$$

with the main difference being that we are concerned with the KL distance, instead of $\ell_2^2$. Given $\tilde{\mathbf{P}}$, user $u$'s assigned cluster $\tilde{z}_u$ is simply given by $\arg\min_k \mathrm{KL}(Q_u \| \mathbf{P}_{:,k})$. Conversely, applying Lemma 3.1, given cluster memberships the cluster center is determined by the cluster average.

**Lemma 3.1.** *For users with assigned cluster $i$, i.e., $\tilde{\mathcal{Z}}_i =: \{u : \tilde{z}_u = i\}$, the following is true:*

$$\frac{1}{|\mathcal{Z}_i|} \sum_{u \in \mathcal{Z}_i} Q_u \in \arg\min_{P \in \Delta(\mathcal{V})} \sum_{u \in \tilde{\mathcal{Z}}_i} \mathrm{KL}(Q_u \| P).$$

Since, we do not have access to $Q_u$, we replace $Q_u$ in Lemma 3.1 and the cluster assignment step with the local Good-Turing [34] estimate $\hat{Q}_u^{\mathrm{gt}}$. Typically, one would use the empirical distribution $\hat{Q}_u$, which is an unbiased estimate of $Q_u$. But, as we will see in Section 5, most real-world distributions are long-tailed and the Good-Turing estimate's approximation of the user distribution does not suffer as heavily from the high-dimensional nature of the problem. We discuss the Good-Turing estimate and analyze its error more formally in Section 4. This simplification means that the cluster center estimator averages the local Good-Turing estimates from each user in the cluster. In theory, we can also estimate the cluster center $P$ in a better way, *e.g.*, by directly running the Good-Turing correction on the cluster level token counts. This is a possible direction of improvement in future works.

To iteratively optimize the objective in Equation 4, we present our algorithm HistogramCluster (see Alg. 3). We start with initial center estimates $\tilde{\mathbf{P}}^{(0)}$, and run $T$ iterations to refine them. In each iteration we peform two steps: (i) *collect* user outputs for each user via UserOutput (see Alg. 1); and (ii) *re-center* clusters using an algorithm, which for now is the NonPrivateCenter (see Alg. 2). Later, we see how we need only update the re-centering algorithm to satisfy user-level privacy contraints. From Lemma 3.1 and Alg. 1, it is not hard to see that NonPrivateCenter which computes the mean of the local estimates of users in a cluster, greedily reduces the objective value in Eq. 4 after each iteration. When the misclustering rate is sufficiently low to begin with, averaging local Good-Turing estimates brings the estimated center closer (in KL divergence) to the true centers that minimize Eq. 4, thereby reducing the number of iterations for convergence.

An extension of the convergence analysis in Balakrishnan et al. [7], gives us an upper bound on the misclustering error of Alg. 3 which goes down exponentially with each iteration of clustering (Theorem 3.2). Similar to most clustering analyses [8, 17, 33], we make suitable conditions on the initialization of cluster centers $\tilde{\mathbf{P}}^{(0)}$ and the separation of true clusters in the user population.

---

**Algorithm 1** UserOutput

**Require:** Centers $\tilde{\mathbf{P}}$, cluster $k$, user $u$.
**Ensure:** User output $(b_u, Q_u)$.
1: $b_u \leftarrow \mathbb{1}(\arg\min_j \mathrm{KL}(\hat{Q}_u^{\mathrm{gt}} \| \tilde{\mathbf{P}}_{:,j}) = k)$
2: $\hat{Q}_u \leftarrow b_u \cdot \hat{Q}_u^{\mathrm{gt}} + (1 - b_u) \cdot \mathbf{0}_d$
3: Return $(b_u, \hat{Q}_u)$.

---

**Algorithm 2** NonPrivateCenter

**Require:** User data $\mathcal{A} =: \{(b_u, \hat{Q}_u)\}$.
**Ensure:** Center $\tilde{P}$.
1: Return $\tilde{P} \leftarrow \sum_u \hat{Q}_u / \sum_u b_u$.

---

**Algorithm 3** HistogramCluster

**Require:** Set of users $\mathcal{U}$, initial centers $\tilde{\mathbf{P}}^{(0)}$, iterations $T$.
**Ensure:** Estimates of cluster memberships $\{\tilde{z}_u\}_u$ and centers $\tilde{\mathbf{P}}$.
1: Initialize $t \leftarrow 1$.
2: **while** $t \leq T$ **do**
3:    **for** $j = 1$ to $K$ **do**
4:       Collect: $\mathcal{A}_k^{(t)} \leftarrow \{\mathrm{UserOutput}(\tilde{\mathbf{P}}^{(t-1)}, k, u)\}_u$
5:       Re-center: $\tilde{\mathbf{P}}_{:,j}^{(t)} \leftarrow (\mathrm{Non})\mathrm{PrivateCenter}(\mathcal{A}_k^{(t)})$
6:    **end for**
7:    $t \leftarrow t + 1$
8: **end while**
9: Return: $\{\tilde{z}_u^{(T)}\}_{u \in \mathcal{U}}$, $\tilde{\mathbf{P}}^{(T)}$.

---

**Theorem 3.2** (Alg. 3 convergence). *For the model in Section 4, if $\forall c, v$, $n_c = \Theta(K^2)$, $P_c[v] = \Omega(1)$, centers are sufficiently separated, i.e., $\Delta =: \min_{i \neq j} \mathrm{KL}(P_i \| P_j) = \Omega(k^2 + k^3 d/n)$, and $\lambda =: \max_{i \neq j} \mathrm{KL}(P_i \| P_j)/\Delta$. Given an initialization with assignment error $O(\sqrt{1/\lambda})$ for any cluster, after $t = \log(|\mathcal{U}|)$ iterations, w.h.p. the assignment error is $O(\exp(-\Delta(\alpha + 1)))$.*

## 3.3 Learning clusters with user-level privacy

First, we note that it suffices to compute all the cluster center estimates in the usual model of DP to ensure that the personalized estimates are differentially private in the billboard model. This is true since the assigment of the user to a cluster center and finetuning can be done locally by each user. Thus, if we ensure that Alg. 2 applied to all the current clusters is differentially private in each iteration we can then bound the total privacy leak across iterations via advanced composition for DP algorithms. In our algorithm PrivateCenter (Alg. 4) we first use the Laplace mechanism in step 1 to privately compute the number of users in the cluster. Then, we use the Gaussian mechanism in step 2 to get an initial estimate of the center since all local Good-Turing estimates are distributions in $\Delta(\mathcal{V})$ and thus bounded in $\ell_2$. This initial estimate $B$ is used to identify the quantile $B \pm c\sqrt{B/b}$, where the mean lies with high probability [69], so that we can then get a refined estimate in step 3 when we add noise (with std. deviation scaling with quantile interval $c$). This adaptive clipping procedure is fairly common for practical and private estimation [4, 12]. Finally, to ensure the noised output is still a distribution contained in $\Delta(\mathcal{V})$, we use $\ell_2$ projection, $\Pi_{\Delta(\mathcal{V})}(P) =: \arg\min_{Q \in \Delta(\mathcal{V})} \|Q - P\|_2^2$. In practice, we find that projection with $\ell_2$ is comparable with a KL projection.

**Privacy preserving noisy sum.** As can be seen from the description of PrivateCenter, it only requires being able to compute a noisy sum over a subset of users. In the federated setting implementing this summation as part of algorithm HistogramCluster may seem to require the server to know the (private) cluster membership of each user. However, this can be avoided by computing each such sum as a sum over all the users, where a user submits value $0$ if they do not belong to the cluster, *i.e.,* $b_u = 0$, and $b_u = 1$ otherwise (Alg. 1). Similarly, Alg. 1 also emits the true local estimate if it is part of the cluster and $\mathbf{0}_d$ (vector of $d$ 0s) otherwise. Privacy-preserving computation of a noisy sum over all the users is one of key primitives in FL with a number of known implementations, *e.g.,* Bonawitz et al. [13]. Crucially for the privacy analysis, while each user now participates in the sum computations for every cluster, the inclusion/deletion of a single user can only affect the sums for a single cluster.

---

**Algorithm 4** PrivateCenter

**Require:** Data collection $\{(b_u, \widehat{Q}_u)\}$, privacy parameter $\rho$, clipping parameter $c$, projection operator $\Pi_{\Delta(\mathcal{V})}$.
**Ensure:** Private center $\tilde{P}$.
1: $b \leftarrow \sum_u b_u + \mathrm{Lap}\left(\sqrt{3/2\rho}\right)$
2: $B \leftarrow \frac{1}{b}\left(\sum_u \widehat{Q}_u + \mathcal{N}(0, 3/2\rho \mathbf{I}_d)\right)$
3: $\Sigma \leftarrow 6c^2/b\rho \cdot \mathrm{diag}(B)$
4: $\tilde{P} \leftarrow \frac{1}{b}\left(\sum_u \bar{Q}_u + \mathcal{N}(0, \Sigma)\right)$,
   $$\bar{Q}_u =: \mathrm{Clip}\,[Q_u]_{B-c\sqrt{\frac{B}{b}}}^{B+c\sqrt{\frac{B}{b}}}$$
5: $\tilde{P} \leftarrow \Pi_{\Delta(\mathcal{V})}\tilde{P}$
6: Return $\tilde{P}$

---

**Algorithm 5** PrivateInit

**Require:** Center $Q_0$, meta-dataset $\mathcal{S}$, # clusters $K$, privacy parameter $\rho$, sampling parameter $\alpha$, clipping threshold $c$.
**Ensure:** Matrix of initial centers $\tilde{P}^{(0)} \in \mathbb{R}^{d \times k}$.
1: Sample $K^2$ candidates: $\mathcal{C} \overset{\mathrm{iid}}{\sim} \mathrm{Dir}(\alpha Q_0)^{K^2}$.
2: Initialize set : $\mathcal{I} \leftarrow \{\}$ and $n \leftarrow |\mathcal{Q}|$.
3: **while** $|\mathcal{I}| < K$ **do**
4:     $\mathcal{I} \leftarrow \mathcal{I} \cup Q$ where we sample $Q$ from $\mathcal{C}$ with,
       $$\Pr(Q) \propto \exp\left(\frac{\sqrt{8\rho}}{2b\sqrt{K}} \sum_{\widehat{Q}_u \in \mathcal{S}} \ell(\widehat{Q}_u, Q)\right), \text{where } \ell(\widehat{Q}_u, Q)$$
       $$=: \mathrm{Clip}\left[\min_{P \in \mathcal{I}} \mathrm{KL}(\widehat{Q}_u \| P) - \min_{P \in \mathcal{I} \cup \{Q\}} \mathrm{KL}(\widehat{Q}_u \| P)\right]_0^c$$
5: **end while**
6: Return $\tilde{P}^{(0)}$ with columns as entries in $\mathcal{I}$.

---

### 3.3.1 Private initialization for our clustering algorithm

The performance of Alg. 3 is crucially determined by the proximity of initialized centers to each of the true centers (Theorem 3.2), and more critical for higher private noisy tolerance. In PrivateInit (Alg. 5), we provide a private data-dependent initialization technique, that takes as input a private center $Q_0$ (private FedAvg), and samples $K^2$ points around it as an initial candidate set. Then, it chooses $K$ points iteratively from this set by sampling via the exponential mechanism. The utility function for each candidate $Q$ is determined by the average reduction in the clustering objective when adding $Q$ into the current set of initializations $\mathcal{I}$, vs. not. The privacy budget $\rho$ controls the temperature, with lower budget enforcing higher smoothing. In the non-private case, we replace the candidate set with the set of empirical user estimates, and select a high value of $\rho$. This iterative procedure of selecting candidates with highest regret of omission at each step is meant to ensure that each $\widehat{Q}_u$ is reasonably close to some initial center (akin to [5]). We remark that the execution of the sampling step does not lend itself easily for implementation in the distributed setting. In practice, one would need to use a small dataset collected centrally for this.

## 3.4 Putting it all together: Algorithm for private and personalized frequency estimation

In Alg. 6 we present our complete algorithm. We first compute a private center of all user histograms using Alg. 4 and privacy parameter $\rho/3$. We pass this to Alg. 5 with privacy parameter $\rho/3$ to get initial centers for Alg. 3, where the re-centering step applies Alg. 4 with privacy parameter $\rho/3T$. The privately learned cluster centers from Alg. 6 are then finetuned for each user locally, using Eq. 3.

**Theorem 3.3** (End-to-end privacy guarantee). *Algorithms 4, 5 are both user-level $\rho$-zCJDP. Our end-to-end private and personalized estimation algorithm is $(\rho + 2\sqrt{\rho \log(1/\delta)}, \delta)$-JDP, $\forall \delta > 0$.*

---

**Algorithm 6** EndToEnd Algorithm

---

**Require:** Set of users $\mathcal{U}$, dataset $\mathcal{S}$, number of clusters $K$, number of clustering iterations $T$, finetuning parameter $\alpha$, sampling parameter $\omega$, privacy parameter $\rho$, clipping parameter $c$, projection operator $\Pi_{\Delta(\mathcal{V})}$.
**Ensure:** Private and personalized estimates $\widehat{Q}_u$ for each $u \in \mathcal{U}$.
1: $Q_0 \leftarrow \text{PrivateCenter}(\mathcal{U}, \rho, c, \Pi_{\Delta(\mathcal{V})})$   *# Estimate global center*
2: $\tilde{\mathbf{P}}^{(0)} \leftarrow$ Algorithm $\text{PrivateInit}(Q_0, \mathcal{S}, K, \rho, \omega, c)$   *# Estimate initial cluster centers*
3: $\{\tilde{z}_u\}_u$ and centers $\tilde{\mathbf{P}} \leftarrow$ Algorithm $\text{HistogramCluster}(\mathcal{U}, \tilde{\mathbf{P}}^{(0)}, T)$   *# Estimate cluster memberships*
4: $\forall u: \quad \tilde{Q}_u \leftarrow \text{FT}(\tilde{\mathbf{P}}_{\tilde{z}_u}, \alpha)$   *# Local finetuning using Eq. 3*
5: Return $\{\tilde{Q}_u : u \in \mathcal{U}\}$.

---

## 3.5 Extending our private clustering algorithm to the size-heterogeneous setting

When users vary in size, we need to weigh their local estimates while estimating the global center or the cluster center in the re-centering step of Alg. 3. For user $u$, the weight $w_u =: (1/\sigma_u^2)/\sum_{u \in \mathcal{Z}_c}(1/\sigma_u^2)$. Here $\sigma_u^2$ is the variance of the user's local estimate, *i.e.*, $\text{Var}[\widehat{Q}_u]$ (from Lemma 3.4). When $P_{z_u}[v]$ is known this weighted estimate is optimal under $\ell_2$ [37]. Since $P_{z_u}[v]$ is unknown we replace $P_{z_u}[v]$ in Lemma 3.4 with a uniform average of $\widehat{Q}_u$ from heavier users, following Cummings et al. [21].

**Lemma 3.4** (Variance of $\widehat{Q}_u$). *For user $u$, $\text{Var}[\widehat{Q}_u[v]] = P_{z_u}[v]/\alpha+1(1 - 1/m_u) + P_{z_u}[v](1-P_{z_u}[v])/m_u$.*

In the private setting though this approach cannot be used directly since the sensitivity of the weighted mean estimate is too high for users with a lot of data, and more so for small sized clusters. Cummings et al. [21] propose an algorithm to estimate the private heterogeneous mean of user data from different Bernoulli distributions, but their approach does not directly transfer to our setting due to the high-dimensional nature of our user means and the relatively poor sensitivity of Alg. 3 to errors in the estimation of cluster centers at each iteration. Thus we consider a two-stage approach, split across data-rich and data-poor users. In the first stage we privately cluster users with sufficiently large datasets applying Alg. 3, and treating them as size-homogeneous data-rich users. Once cluster centers are learned privately, using Alg. 1, each data-poor user assigns itself to the closest center in KL divergence. Finally for each cluster, we apply Algorithm 2 from Cummings et al. [21] that re-centers each cluster privately, based on the newly added users.

## 4 Formal analysis in a stylized data model

In Section 3, we presented a clustering based iterative algorithm (Alg. 3) for our frequency estimation problem introduced in Section 2, wherein we made several key design choices. In particular, we used average of Good-Turing estimators to estimate the cluster center, and used the update in Equation 3 to locally finetune the learnt cluster centers. Now, we analyze these algorithms in a stylized model.

**Bayesian Model.** Each user $u$ belongs to a cluster $z_u \in [K]$ with cluster center $P_{z_u}$. The user histogram $Q_u \sim \text{Dir}(\alpha P_{z_u})$, and the user dataset $\mathcal{S}_u \sim Q_u^{m_u}$ is sampled *i.i.d.* from $Q_u$. We use $\mathcal{Z}_c =: \{u : z_u = c\}$ to denote the set of users from cluster $c$, and $n_c =: |\mathcal{Z}_c|$ is the number of users in $c$. Higher value of $\alpha$ implies more concentrated clusters since $\text{Var}(Q_u[v]) = O(P_{z_u}[v]/1+\alpha)$, and as $\alpha \to \infty$, $Q_u \to P_{z_u}$ in weak topology [28]. Please note that the Dirichlet assumption is mainly for simplicity and our results (*e.g.*, Theorem 4.3) only require each cluster's user distribution to be exponentially concentrated along each token, *i.e.*, $\Pr(|Q_u[v] - P_{z_u}[v]| \geq t) = \mathcal{O}(e^{-t})$.

**Purely local learning.** We present two local estimators that estimate $Q_u$: empirical $\widehat{Q}_u$. and Good-Turing [30] $\widehat{Q}_u^{\text{gt}}$. The naïve estimate $\widehat{Q}_u$ is the average of user data in $\mathcal{S}_u$. Next, we define the

Good-Turing estimator. Let the frequency of token $v$ in user's dataset $\mathcal{S}_u$ (of size $m_u$) be $\mathrm{cnt}_{u,v}$. We denote the local frequency of the count $j$ as $\phi_{u,j} =: \sum_{v \in \mathcal{V}} \mathbb{1}(\mathrm{cnt}_{u,v} = j)$, *i.e.*, the number of token in $\mathcal{S}_u$ with count $j$. Following Orlitsky and Suresh [63], for a token $v$ that appears $j = \mathrm{cnt}_{u,v}$ times, the Good-Turing estimator for $Q_u[v]$ is given by:

$$\widehat{Q}_u^{\mathrm{gt}}[v] =: \begin{cases} \frac{j}{m_u N_u} & \text{if } j > \phi_{u,j+1}, \\ \frac{j+1}{m_u N_u} \cdot \frac{\phi_{u,j+1}+1}{\phi_{u,j}} & \text{otherwise},\end{cases} \tag{5}$$

where $N_u$ is the normalization factor so that $\sum_v \widehat{Q}_u^{\mathrm{gt}}[v] = 1$. Let $\mathcal{E}_u$ be the class of natural estimators that assign the same probability to tokens with equal counts in $\mathcal{S}_u$. Then, following Lemma 6 in Orlitsky and Suresh [63] and Theorem 2 in Acharya et al. [2] we conclude that with respect to $\mathcal{E}_u$, the Good-Turing estimate $\widehat{Q}_u^{\mathrm{gt}}$ has a worst case suboptimality gap of $\tilde{O}(\min(\sqrt{1/m}, d/m))$ (see Lemma 4.1 for the full statement). The key point to note here is that when the user dataset is small, we do not suffer from the vocabulary size $d$, unlike the empirical estimate $\widehat{Q}_u$. This is mainly because Good-Turing more accurately estimates the probability of unseen words [56].

**Lemma 4.1.** *($\widehat{Q}_u^{\mathrm{gt}}$ suboptimality gap) For any $Q_u$, the suboptimality gap of $\widehat{Q}_u^{\mathrm{gt}}$ with respect to $\mathcal{E}_u$ is $\mathbb{E}[\mathrm{KL}(Q_u \| \widehat{Q}_u^{\mathrm{gt}})] - \min_{\tilde{A} \in \mathcal{E}_u} \mathbb{E}[\mathrm{KL}(Q_u \| \tilde{A})] = \tilde{O}(\min(\sqrt{1/m}, d/m))$, that matches minimax rates.*

**Bayes optimal finetuning when the cluster center is given.** When $P_{z_u}$ is known, FT in Eq. 3 is Bayes optimal in KL and when only an estimate of the center is known, the error of the "plug-in" estimate scales linearly with the error in the center's estimate (Theorem 4.2).

**Theorem 4.2** (Bayes optimal local learning). *Given $P_{z_u}, \alpha$, the estimator $\mathrm{FT}(P_{z_u}, \widehat{Q}_u)$ is Bayes optimal in KL divergence, for the Dirichlet prior $\mathrm{Dir}(\alpha P_{z_u})$. When $\tilde{P}_{z_u}$ is the estimated cluster center, and $\tilde{Q}_u^{\mathrm{opt}}$ is the Bayes optimal estimate of $Q_u$, then $\mathrm{KL}(\tilde{Q}_u^{\mathrm{opt}} \| \mathrm{FT}(\tilde{P}_{z_u}, \widehat{Q}_u)) \leq \alpha/\alpha+m_u \, \mathrm{KL}(P_{z_u} \| \tilde{P}_{z_u})$.*

**Estimating the cluster center given cluster members.** Given all users in a cluster $c$, one estimate of the cluster center $P_c$ is the solution to the maximum log-likelihood objective in Eq. 6, which is typically solved with the FedAvg [49] algorithm. We denote this estimate as $\tilde{P}_c^{\mathrm{fa}}$ and is a simple weighted average of empirical estimates (see proof in Appendix E). Similarly, for user $u$ in cluster $c$, the personalized output from the PFL algorithm FedAvg+FT is given by $\mathrm{FT}(\tilde{P}_c^{\mathrm{fa}}, \widehat{Q}_u)$.

$$\tilde{P}_c^{\mathrm{fa}} =: \sum_{u \in \mathcal{Z}_c} \left( m_u/\sum_{u' \in \mathcal{Z}_c} m_{u'} \right) \cdot \widehat{Q}_u \ \in \ \arg\max_{Q \in \Delta(\mathcal{V})} \frac{1}{n_c} \sum_{u \in \mathcal{Z}_c} \log \Pr(\mathcal{S}_u \mid Q) \qquad \text{(FedAvg)} \tag{6}$$

Our algorithms (Alg. 6, Alg. 2) use a different estimator. In the size-homogenous case, to estimate the cluster center we average the local Good-Turing estimates $\widehat{Q}_u^{\mathrm{gt}}$ for all users in the cluster:

$$\tilde{P}_c^{\mathrm{gt}} =: \frac{1}{n_c} \sum_{u \in \mathcal{Z}_c} \widehat{Q}_u^{\mathrm{gt}} \qquad \text{(Good-Turing based; ours).} \tag{7}$$

To analyze the accuracy guarantees of our estimate $\tilde{P}_c^{\mathrm{gt}}$, we first define the class of competing estimators for it. Let $\bar{\mathrm{cnt}}_v =: \lfloor \sum_{u \in \mathcal{Z}} \mathrm{cnt}_{u,v}/\sum_u m_u \rfloor$ be the average count of token $v$ across all users in a cluster. We define $\bar{\mathcal{E}}$ to be the class of estimators that assign the same mass to tokens $v_1, v_2$ if $\bar{\mathrm{cnt}}_{v_1} = \bar{\mathrm{cnt}}_{v_2}$. Such a class is natural for estimators that rely on aggregated statistics, and includes the Good-Turing correction applied to cluster level counts. In Theorem 4.3, for any cluster center $P_c$, we compare the suboptimality gap in KL divergence for our estimate $\tilde{P}_c^{\mathrm{gt}}$ in Eq. 7 with FedAvg.

**Theorem 4.3** ($\tilde{P}_c^{\mathrm{gt}}$ in Eq. 7 has lower suboptimality gap than FedAvg). *For the competetive class of estimators with the same average count, i.e., $\bar{\mathcal{E}}$ (defined above), and suboptimality gap from Lemma 4.1, the suboptimality gap for $\tilde{P}^{\mathrm{gt}}$ in Eq. 7, w.r.t. $\bar{\mathcal{E}}$ is $\tilde{O}(\alpha+m/\alpha+1 \cdot \min(\sqrt{1/m}, d/m))$. The suboptimality gap over $\bar{\mathcal{E}}$ for FedAvg $\tilde{P}^{\mathrm{fa}}$ is $\tilde{\Omega}(d(\alpha+m)/m(\alpha+1))$. $\tilde{O}, \tilde{\Omega}$ hides polylog factors in $m, n$.*

When $m$ is small and comparable to the intra-cluster std. deviation, i.e., $m/\alpha+1 = O(1)$, then the suboptimality gap for our estimator does not suffer from large dimension $d$. On the other hand, FedAvg guarantees are much weaker and degrade with dimension for small user datasets (common in practice). From, Theorem 4.2, we also conclude that our final estimate for $Q_u$, given by the finetuned center: $\mathrm{FT}(\tilde{P}, \widehat{Q}_u)$ would also have stronger error guarantees than FedAvg+FT.

**Partitioning users into clusters.** In practice, neither the cluster membership $z_u$ nor the center $P_c$ is known. The Bayes optimal estimate for $P_c, z_u$ is the mean of the posterior $\Pr(Q_u \mid \mathcal{S}, \alpha)$

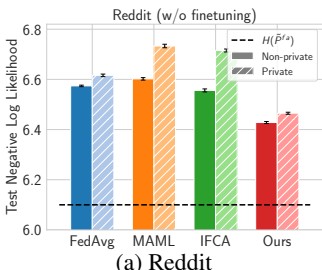
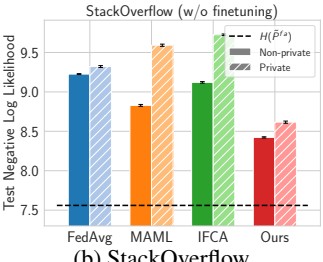
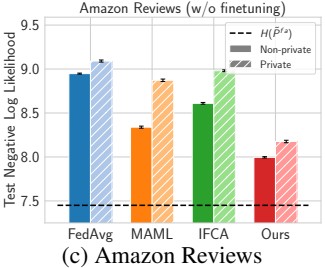

(a) Reddit        (b) StackOverflow        (c) Amazon Reviews

Figure 1: *Performance before finetuning:* We compare the test NLL loss before local personalization (finetuning) for baselines FedAvg, MAML, IFCA with our approach. NLL is uniformly averaged over users and each value is averaged over 50 random runs (error bars indicate 95% confidence intervals).

(see Theorem 4.2 proof), but since the posterior is intractable, we instead compute the maximum likelihood estimate (MLE). Asymptotically, MLE's accuracy is vindicated by Doob's theorem [59], and its variance matches the Cramér-Rao bound [69]. Still, the joint MLE for $P_c, z_u$ is a non-concave maximization problem. Akin to solving the MLE for mixture distributions [61], we maximize an evidence lower bound with Expectation-Maximization (EM) [7]. Since each $Q_u$ belongs to only one cluster, it must be that $z_u \in \arg\min_{c \in [k]} \mathrm{KL}(Q_u \| P_c)$. Thus, the MLE for the cluster centers is given by objective in Eq. 4 from Section 3, which is optimized iteratively by our Algorithm 6.

**Takeaways.** We note: (i) when cluster memberships are known, average of local Good-Turings has a lower suboptimality gap than FedAvg; and (ii) solutions to the clustering objective in Eq. 4 estimates cluster centers for each user; and (iii) if the center estimates are accurate then finetuning them with FT yields estimates that are equally close to the Bayes optimal solution. These findings validate our algorithmic design choices for the clustering and finetuning algorithms in Section 3.

## 5 Empirical evaluation

We empirically evaluate our approach on real-world datasets and present: **(i)** contrary to popular belief [76], we show there are real-world distributions where clustering based algorithms (*e.g.*, ours) significantly outperform the popular PFL baseline FedAvg+FT; **(ii)** our method achieves better privacy-utility tradeoff than private versions of clustering-based baselines, and the gap amplifies in the size-heterogeneous case; and **(iii)** we show our method's performance improvements can be largely attributed to algorithmic design choices influenced by our data model and analysis in Section 4.

**Datasets.** We evaluate methods on three real-world datasets: Reddit [16], StackOverflow [6], and Amazon Reviews [62]. For Reddit, we use the NLTK tokenizer [11] with a vocabulary of size 10k tokens, and for the other two datasets, we use the Huggingface (bert-case-uncased) tokenizer [74] with a vocabulary size of 32k tokens. For more details on datasets and hyperparameters see Appendix C.

**Baselines.** We evaluate two baselines that learn a single global model: FedAvg [57] and MAML [29]; and a clustering-based approach which learns multiple models, one for each cluster: IFCA [33]/HypCluster [54]. Given a single global model (or a cluster level model), each user can finetune this locally in different ways: full batch gradient descent (GD) initialized at global model, RTFA [19], or with our problem specific method in Eq. 3. We compare these on FedAvg model

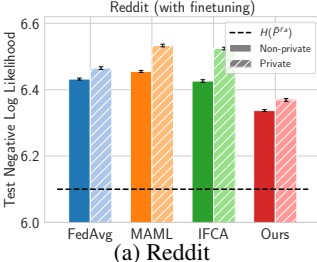
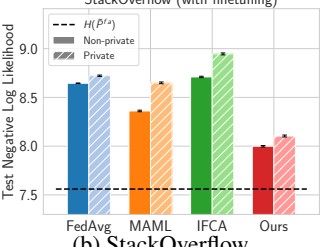
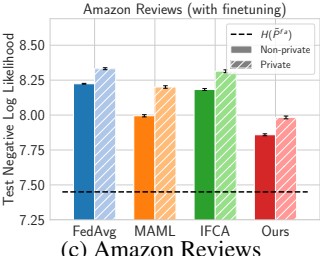

(a) Reddit        (b) StackOverflow        (c) Amazon Reviews

Figure 2: *Performance after finetuning:* In the size-homogeneous (a-c), and size-heterogenenous (d-f) settings, we compare the test NLL loss for baselines FedAvg+FT, MAML+FT, IFCA+FT with our Alg. 3+FT, where FT is implemented by Eq. 3. Uniformly averaged over users, each value is averaged over 50 random runs (error bars indicate 95% confidence intervals).

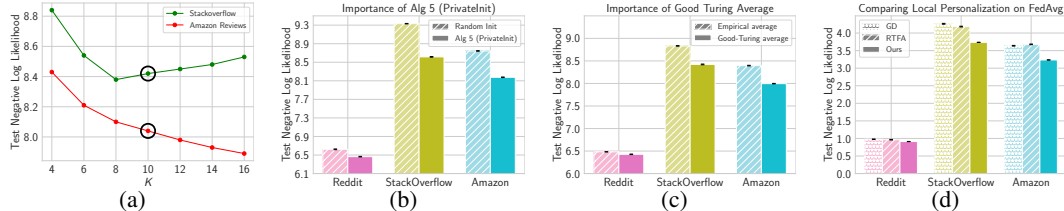

Figure 3: *Algorithmic design choices:* We evaluate test NLL for Alg. 3 as we: (a) vary the number of clusters $K$; (b) use PrivateInit or randomly initialize cluster centers; and (c) use average of Good-Turing or empirical average to estimate cluster centers. In (d) we evaluate different finetuning methods applied to the FedAvg model.

(Figure 3(d)), and fix the best one (ours) as the local finetuning approach for all algorithms. In the private setting, we adapt the baselines by using standard techniques [1, 25] that privatize $l_2$ bounded gradients for MAML/IFCA, and $l_1$ bounded probability distributions for FedAvg.

**Evaluation metric.** For distribution $Q_u$, estimate $\tilde{Q}_u$, $\mathrm{KL}(Q_u, \tilde{Q}_u) = \mathrm{NLL}(Q_u, \tilde{Q}_u) - \mathrm{H}(Q_u)$, where $\mathrm{NLL}(Q_u, \tilde{Q}_u) =: -\sum_{w \in \mathcal{V}} Q_{u,w} \log Q_{u,w}$ is the negative log-likelihood loss, and $\mathrm{H}(Q_u)$ is the entropy of $Q_u$. Following language modeling [3, 38], we use the test set of the user to get an unbiased estimate of the NLL loss, by replacing $Q_u$ with the empirical distribution on the test set. Since the test set is insufficient to get an unbiased estimate of $\mathrm{H}(Q_u)$, as a reference point, we instead report the entropy of the global center or FedAvg estimate $\mathrm{H}(\tilde{P}^{fa})$, which is a rough estimate of the "hardness" of estimating the user distribution. We refer to this difference in test NLL and $\mathrm{H}(\tilde{P}^{fa})$ as *NLL sub-optimality gap*. In the private case, we ensure the algorithms satisfy $(15, 10^{-10})$-JDP.

**Our approach significantly reduces test NLL in private and non-private settings.** In Figure 1 we report a significant reduction in test NLL sub-optimality gap even before local finetuning, by atleast 25-45% in the non-private setting and upto 50% in the private case. Non-privately MAML/IFCA perform better than FedAvg on StackOverflow and Amazon, but their private versions perform similar or worse than the private FedAvg. This is because the gradient-based optimization in MAML and IFCA can incur very high per-iteration privacy overheads which only accumulates more for the latter that additionally suffers from poor convergence due to imperfect cluster initialization. On the other hand, our method achieves a privacy-utility tradeoff that is comparable to the gradient-free FedAvg, and converges in fewer iterations when clustering is initialized with centers from Alg. 5. In Figure 2 we compare the performance after we finetune the global/cluster-level estimates locally for each user (using Eq. 3), and verify that the relative gains remain consistent with pre-finetuning. In particular we note that finetuning global FedAvg/MAML models does not do better than finetuning cluster-level models, vindicating the presence of concentrated subpopulations in an otherwise highly heterogeneous real-world user distribution.

**Algorithmic design decisions influenced by our model in Section 4.** In Section 4 we identified the Good-Turing based estimator (Eq. 7) to suffer less from vocabulary size, compared to FedAvg (Theorem 4.3), and in practice too we observe a big improvement in the test NLL loss of Alg. 3 when it uses Good-Turing as the local estimate, vs. empirical (Figure 3(c)). The relative gap is particularly wider on the large vocabulary datasets Amazon and StackOverflow. In Figure 3(b), we show that our data-dependent private initialization (Alg. 5), that is also based on our mixture of Dirichlet model, plays a crucial role in lowering the final test loss, and we attribute this gain to a significant reduction in clustering iterations (from 200 to 20), thereby reducing privacy overhead. In Figure 3(d) we note that the Bayes optimal finetuning algorithm in our data model (Eq. 3) does better than typical gradient-based (GD) or proximal term based (RTFA) approaches. Finally, in Figure 3(a), we plot test NLL as we vary the number of clusters assumed by Alg. 3. We note that for runs on all datasets we choose $K=10$ which was found to be optimal on the Reddit validation set (see Appendix C), even though it is clearly (slightly) suboptimal on the other two, suggesting that Alg. 3's performance is not too sensitive to the choice of $K$ practice.

**Size-heterogeneous case.** In Figure 4 plot test NLL in the size-heterogeneous case where our algorithm's gains ($> 40\%$ error reduction) over single global model baselines FedAvg and MAML are even more pronounced than the size-homogeneous results in Figures 1, 2. For the better suited clustering baseline IFCA, the privacy utility tradeoff worsens (compared to size-homogenous) due to poorer higher sensitivity of gradient estimates when dataset sizes vary. On the other hand, our two-stage approach reduces privacy noise in the clustering stage by partitioning users into different stages and still improves performance for data scarce users in the second stage by incorporating them

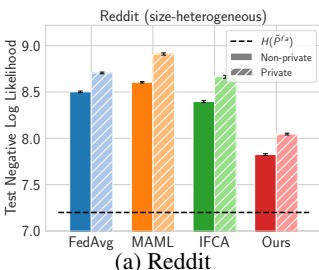
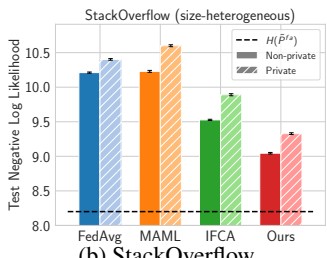
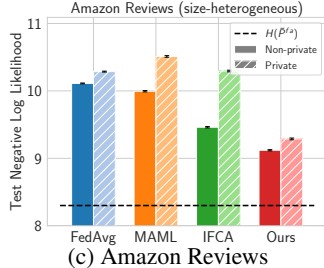

| (a) Reddit | (b) StackOverflow | (c) Amazon Reviews |

Figure 4: *Size-heterogenous setting:* Comparison of test loss for baselines FedAvg+FT, MAML+FT, IFCA+FT with our Alg. 3+FT, in the size-heterogeneous setting where each user's dataset vastly differs in size. Uniformly averaged over users, each value is averaged over 50 random runs (error bars indicate $95\%$ confidence intervals).

in the re-centering of clusters from previous stage. Compared to IFCA we observe $> 50\%$ reduction in test NLL suboptimality gap for private estimation on Amazon Reviews.

## 6 Related work

**Private and personalized federated learning.** Multiple works propose and analyze clustering based algorithms that learn a diverse set of models for heterogeneous user distributions [26, 33, 35, 54, 55, 66, 73, 77]. With the exception of [66] that first learns a global model and then uses it to partition users based on their losses, most works learn diverse models from scratch using gradient based algorithms, as done by FedAvg [57]. The key difference being that they first partition clients based on their loss [33] or gradients [72] and then use the same gradients to update different models, one for each cluster. Additionally, their analysis also holds mainly for smooth/strongly convex loss functions, for *e.g.*, least squares [33, 54]. This is a natural and promising approach but we are not aware of practical results showing that it can improve on the more direct combination of FedAvg and FT [19, 76]. In fact, Wu et al. [76] raise concerns about mode collapse with clustering iterations. Contrary to the works above, we focus on a non-gradient based approach specifically for frequency estimation in KL divergence, which can be ill-conditioned in practice, and also provide an algorithm for the more challenging size-heterogeneous setting. As we show in our work, this problem requires solutions that are different from the ones proposed for general loss families in machine learning. Moreover, unlike the above, we provide privacy guarantees for our clustering based personalization. For more discussion on related works please see Appendix B

**Distribution estimation in KL divergence.** A multitude of works on mixture of Gaussians, and mixed linear regression propose and analyze distribution estimation algorithms [9, 22, 45, 61], but their guarantees are mainly for parametric estimation errors in $\ell_1/\ell_2$ metric. In contrast, we are concerned with histogram estimation in KL divergence which presents interesting challenges since this is not a proper distance metric (*e.g.*, does not satisfy triangle inequality). On the other hand, [30, 34, 64] study Good-Turing estimators and give estimation error guarantees in KL divergence for categorical distributions over a fixed alphabet. Our work extends estimators of Acharya et al. [2], Orlitsky and Suresh [63] to the federated setting where the goal is to estimate a full population of distributions that share a latent structure. Relevant to our objective and metric is [17] that analyzes guarantees for information theoretic clustering. Their analysis shows that one can adapt analysis for other metrics (*e.g.*, Hellinger) to obtain worst-case approximation guarantees. In contrast, we investigate practical and private algorithms for the federated setting.

## 7 Conclusion

We introduce the problem of private and personalized frequency estimation in KL divergence. For this, we propose an iterative algorithm that privately learns clusters in the population of all user frequencies. Each user in a cluster locally finetunes their corresponding cluster center to produce personalized and private frequency estimates with formal joint DP guarantees. We improve the privacy-utility tradeoff of our algorithm by proposing a novel data-dependent private initialization for clustering that empirically reduces number of clustering iterations. We also present a two-stage version of our approach to separately handle the harder size-heterogenous setting. In a Bayesian model where the user distributions are distributed as Dirichlets around well-separated centers, we reason about different collaborative and local estimators, and provide formal guarantees for some of our algorithmic design choices, like Good-Turing estimators, and the choice of the local finetuning algorithm. Empirically, we test our algorithm on three real-world datasets and show a significant reduction in test NLL, by 25-45% in the non-private setting and upto $50\%$ in the private case.

**Acknowledgements**

Part of this work was done when AS was an intern at Apple, hosted by VF and KT. AS would like to thank Virginia Smith, Moshe Shenfeld, Hilal Asi, Abhishek Shetty, Aadirupa Saha, Steven Wu, Shengyuan Hu, Pratiksha Thaker, Tian Li, and Don Dennis for helfpful feedback and discussions. AS also thanks JP Morgan AI PhD Fellowship for their generous support.

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

# Appendices

## A    Limitations of our work

As a first step towards language modeling we only focus on privately estimating the marginal token distribution for each user, and an interesting future direction would be to extend this to conditional next-token distributions. Nevertheless, we find that existing algorithms in the federated learning literature when applied to our problem setting perform suboptimally, and require problem specific interventions we introduce, like Algorithms 3, 5, and Good-Turing based estimators. Our current work also does not handle a federated version of the private initialization algorithm we propose. We suggest handling the initialization at the central node that maintains a small dataset of clients. This is a known limitation for sampling from the exponential mechanism in a federated setting. At the same time we would like to point out, that all our other algorithms, including private ones can be implemented in the federated setting.

## B    Additional related work

In our work we consider joint, user-level privacy guarantees in the billboard model [39], whereas some prior works at the intersection of privacy and personalization only provide weaker *record-level* privacy guarantees [40, 51]. While other works [50, 58] study the notion of user-level privacy in federated setups, they are mainly concerned with learning a single model. Cummings et al. [21] also study the problem of private heterogeneous mean estimation but do not consider the clustered setting, or the subtleties of estimatation in KL divergence. More recent works (*e.g.*, [42, 43]) propose personalization algorithms that are user-level private, but are mainly tailored for settings where the different models for each user share representations or lie in a hidden low rank subspace. In contrast, we explore another latent structure more suited for histogram estimation, *i.e.*, one where histograms need only be concentrated around the vertices of a low-dimensional polytope. Other works [10, 41, 75] propose gradient-based algorithms for private multi-task learning where they learn different models for each user (task), regularized using inter-task prior relationship matrices. In contrast, our work does not assume any such prior knowledge.

## C    Additional details for empirical evaluation

**Datasets.** We evaluate methods on three real-world datasets: Reddit [16], StackOverflow [6], and Amazon Reviews [62]. For the Reddit dataset, we use the NLTK tokenizer [11] with a vocabulary of size 10k tokens, and use the Huggingface (bert-case-uncased) tokenizer [74] with a size of 32k tokens for the other two. We have $\approx$ 10k users in Reddit, each with at least 1k tokens. For StackOverflow and Amazon Reviews we have $\approx$ 19k and 50k clients respectively, each with 500 tokens For all datasets, we partition the data for each client into $60 : 40$ train/test splits. Additionally, we set aside $5\%$ of users in each, as a validation, to tune cluster count $K$, privacy clip bound $c$, etc. In the private setting, we adapt the baselines to the private setting by using standard techniques [1, 25] to privatize $l_2$ bounded gradients for MAML/IFCA, and $l_1$ bounded probability distributions for FedAvg, so as to ensure all algorithms achieve JDP-$(15, 10^{-10})$. For more details see Appendix C.

**Hyperparameters.** We validate hyperparameters for our algorithms and baselines using a hold out validation set of users. For clustering we use $K = 10$, as validated by Figure 5, and fix this for all datasets and clustering baselines. We run clustering for $T = 50$ iterations non-privately and $T = 20$ iterations privately on all datasets. We tune the finetuning parameter $\lambda$ by sweeping across $\{0.05, 0.1, 0.15, 0.2, 0.25, 0.3, 0.35, 0.4, 0.45, 0.5\}$ and find $\lambda = 0.25$, $\lambda = 0.15$ and $\lambda = 0.1$ to be optimal on Reddit, StackOverflow and Amazon respectively. We train MAML and IFCA using SGD with momentum, with learning rate $0.01$ and momentum parameter $0.9$. For RTFA, we find the proximal regularization parameter of $0.2$ to be optimal on Reddit and StackOverflow, $0.1$ to perform better on Amazon Reviews. For private training, we use a clipping threshold of $c = 0.1$ in Alg. 4, and use $c = 4.0$ for Alg. 4.

**Computational resources.** None of our experiments require very high computational requirements and can be run with one 3090Ti card. All runs can be reproduced in approximately 500 GPU hours.

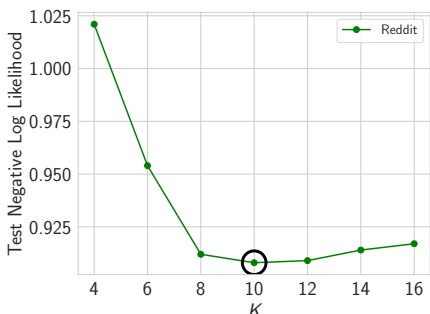

Figure 5: Validating hyperparameter choice of $K = 10$ on Reddit dataset. We use the same value of $K$ for the other two datasets as well.

# D  Useful Lemmas

**Lemma D.1** (Chernoff bound). *If $X \sim Poi(\lambda)$, then for $x \geq \lambda$,*

$$\Pr(X \geq x) \leq \exp\left(-\frac{(x-\lambda)^2}{2x}\right), \tag{8}$$

*and for $x < \lambda$,*

$$\Pr(X \leq x) \leq \exp\left(-\frac{(x-\lambda)^2}{2\lambda}\right). \tag{9}$$

**Lemma D.2** (Bernstein's Inequality). *Let $X_1, X_2, \ldots, X_n$ be $n$ independent zero mean random variables such that with probability $\geq 1 - \epsilon_i$, $|X_i| < M$. Then,*

$$\Pr\left(\left|\sum_i X_i\right| \geq t\right) \leq 2\exp\left(-\frac{t^2}{\sum_i \mathbb{E}[X_i^2] + Mt/3}\right) + \sum_{i=1}^{n} \epsilon_i. \tag{10}$$

*If $t = \sqrt{2\left(\sum_i \mathbb{E}[X_i^2]\right)\log\frac{1}{\delta} + \frac{2}{3}M\log\frac{1}{\delta}}$, then*

$$\Pr\left(\left|\sum_i X_i\right| \geq \sqrt{2\left(\sum_i \mathbb{E}[X_i^2]\right)\log\frac{1}{\delta} + \frac{2}{3}M\log\frac{1}{\delta}}\right) \leq 2\delta + \sum_{i=1}^{n} \epsilon_i. \tag{11}$$

**Lemma D.3** (Banerjee et al. [8]). *The solution for the optimization problem:*

$$\underset{Q}{\arg\min}\ (1-\lambda)\,\mathrm{KL}(P_0\,\|\,Q) + \lambda\,\mathrm{KL}(P_1\,\|\,Q)$$

*is given by $(1-\lambda)P_0 + \lambda P_1$.*

# E  Missing proofs from Section 4

**Lemma E.1.** *($\widehat{Q}_u^{\mathrm{gt}}$ achieves minimax suboptimality gap; restated) For any $Q_u$, the suboptimality gap of $\widehat{Q}_u^{\mathrm{gt}}$ with respect to $\mathcal{E}_u$ is $\mathbb{E}[\mathrm{KL}(Q_u\|\widehat{Q}_u^{\mathrm{gt}})] - \min_{\tilde{A}\in\mathcal{E}} \mathbb{E}[\mathrm{KL}(Q_u\|\tilde{A})] = \tilde{O}(\min(\sqrt{1/m}, d/m))$, that matches minimax rates.*

*Proof.* From Lemma 8 in [63] we know that for the class of natural estimators $\mathcal{E}_u$, the suboptimality gap defined above is given by $\mathbb{E}[\mathrm{KL}(M_k\|\tilde{M}_k)]$, where

$$M_k =: \sum_{v\in\mathcal{V}} \mathbb{1}(\mathrm{cnt}_v = k)Q_u[v], \quad \tilde{M}_k =: \sum_{v\in\mathcal{V}} \mathbb{1}(\mathrm{cnt}_v = k)\tilde{Q}_u[v].$$

We use the definition of $\phi_{u,t}$ from [63], where:

$$\phi_{u,t} = \sum_{v\in\mathcal{V}} \mathbb{1}(\mathrm{cnt}_{u,v} = t) \tag{12}$$

Now, from Theorem 2 in [63] we also know that $\mathrm{KL}(M_k\|\tilde{M}_k) = \tilde{O}(\sqrt{1/m})$ with probability at least $1 - 1/m$ over the randomness in draw of the user dataset $\mathcal{S}_u$. Since their estimate is lower bounded by $1/2m$, we can convert their high probability guarantees to guarantees in expectation. Further, we note that in their proof of Theorem 2, we can easily upper bound the following

$$\sum_t \frac{(M_k - \tilde{M}_k)^2}{\tilde{M}_k} \leq \sum_k \frac{\sqrt{\phi_{u,t}}}{m} \leq \sum_t \frac{\phi_{u,t}}{m} = \frac{d}{m}$$

in the worst case. Thus, in the worst case, $\mathbb{E}[\mathrm{KL}(M_k\|\tilde{M}_k)]$ is upper bounded by the minimum of $\mathcal{O}(d/m)$ and $\tilde{\mathcal{O}}(\sqrt{1/m})$. This completes the proof of Lemma 4.1. $\qquad\square$

**Theorem E.2** (Finetuning is Bayes optimal given center; restated)**.** *If cluster center $P_{z_u}$, and Dirichlet parameter $\alpha$ is known then $\mathrm{FT}(P_{z_u}, \widehat{Q}_u)$ is Bayes optimal in KL divergence.*

*Proof.* We note that the distribution of $\widehat{Q}_u$ given the center $P_{c_u}$ is a Dirichlet-Multinomial distribution. Further, the Dirichlet distribution is a conjugate prior for the Multinomial.

Hence, the posterior distribution for $Q_u \mid \widehat{Q}_u, P_{c_u}$ is a Dirichlet Multinomial with mean: $(\frac{\alpha}{\alpha+m})P_u + (\frac{m}{\alpha+m})\widehat{Q}_u$. To see why, mean is the Bayes optimal estimator, we invoke Lemma D.3, which implies:

$$(1-\lambda)P + \lambda\widehat{Q}_u \in \arg\min_Q \; (1-\lambda)\mathrm{KL}(P\,\|\,Q) + \lambda\,\mathrm{KL}(\widehat{Q}_u\,\|\,Q)$$

Now the Bayes risk of algorithm $\mathfrak{M}$, given the center estimate $P$ is :

$$\mathrm{BayesRisk}(\mathfrak{M}) =: \mathbb{E}_{Q_u \sim \mathrm{Pr}(Q_u|\widehat{Q}_u, P)}\left[\mathrm{KL}(Q_u\|\mathfrak{M}(\widehat{Q}_u))\right]$$

Recursively applying Lemma D.3 on every measurable subspace (under measure defined by the posterior $\mathrm{Pr}(Q_u \mid \widehat{Q}_u, P)$) of the set of token distributions $\Delta(\mathcal{V})$, we conclude that the optimal solution is $\mathbb{E}[Q_u|\widehat{Q}_u, P]$ when $Q_u \sim \mathrm{Pr}(Q_u \mid \widehat{Q}_u, P)$. This concludes our proof. Note that the same proof also applies for Lemma 3.1, but applied iteratively on each element in the summation. $\qquad\square$

**Lemma E.3** (Accuracy of plugging center estimate into finetuning; restated)**.** *Let $\tilde{P}_{z_u}$ be the estimated cluster center for user $u$, and $\tilde{Q}_u^{\mathrm{opt}}$ be the Bayes optimal estimate of $Q_u$. Then,*

$$\mathrm{KL}(\tilde{Q}_u^{\mathrm{opt}}\|\mathrm{FT}(\tilde{P}_{z_u}, \widehat{Q}_u)) \leq \frac{\alpha}{\alpha + m_u}\mathrm{KL}(P_{z_u}\|\tilde{P}_{z_u}).$$

*Proof.* This result is a simple application of Jensen's inequality for KL divergence.

$$\begin{aligned}
&\mathrm{KL}(\tilde{Q}_u^{\mathrm{opt}}\|\mathrm{FT}(\tilde{P}_{z_u}, \widehat{Q}_u)) \\
&= \mathrm{KL}\left(\left(\frac{\alpha}{\alpha + m_u}P_{z_u} + \frac{\alpha}{\alpha + m_u}\widehat{Q}_u\right) \Big\| \left(\frac{\alpha}{\alpha + m_u}\tilde{P}_{z_u} + \frac{\alpha}{\alpha + m_u}\widehat{Q}_u\right)\right) \\
&\leq \frac{\alpha}{\alpha + m_u}\mathrm{KL}(P_{z_u}\|\tilde{P}_{z_u})
\end{aligned}$$

$\qquad\square$

**Lemma E.4** (FedAvg estimate)**.** *The FedAvg model is given by $Q_{\mathrm{fa}} = \frac{1}{n}\sum_{\widehat{Q}_u \in \mathcal{S}} \widehat{Q}_u$.*

*Proof.* The log likelihood objective $\log \mathrm{Pr}(\mathcal{S}_u\|Q)$ is equivalent upto additive constant with the objective $\mathrm{KL}(\widehat{Q}_u\|Q_u)$. We can once again use the result from Lemma D.3 to conclude that the optimal is simply the average of local means. $\qquad\square$

**Theorem E.5** ($\tilde{P}_c^{\mathrm{gt}}$ in Eq. 7 has lower suboptimality gap than FedAvg; restated)**.** *For the competetive class of estimators with the same average count, i.e., $\mathcal{E}$ (defined above), and suboptimality gap from Lemma 4.1, the suboptimality gap for $\tilde{P}^{\mathrm{gt}}$ in Eq. 7, w.r.t. $\bar{\mathcal{E}}$ is $\tilde{O}(\alpha+m/\alpha+1 \cdot \min(\sqrt{1/m}, d/m))$. The suboptimality gap over $\bar{\mathcal{E}}$ for FedAvg $\tilde{P}^{\mathrm{fa}}$ is $\tilde{\Omega}\left(d(\alpha+m)/m(\alpha+1)\right)$. $\tilde{O}, \tilde{\Omega}$ hides polylog factors in $m, n$.*

*Proof.* For notational convenience, we denote the cluster center with $P$ (instead of $P_c$), and the number of users in the cluster as $n$ (instead of $n_c$). To derive bounds on competetive estimation, where the competetive class can only use the average count for each token, we need to first ensure that our Good-Turing based estimator can actively estimate the following quantity:

$$M_t =: \sum_{v \in \mathcal{V}} \mathbb{1}(\lfloor \overline{\mathrm{cnt}}_v \rfloor = t) \, P_v \quad \text{and} \quad F_t =: \sum_{v \in \mathcal{V}} \mathbb{1}(\lfloor \overline{\mathrm{cnt}}_v \rfloor = t) \, \tilde{P}_v^{\mathrm{gt}} \tag{13}$$

Similarly, the local equivalent of $M_t$ for each user is defined as:

$$M_{u,t} = \sum_{v \in \mathcal{V}} \mathbb{1}(\mathrm{cnt}_{u,v} = t) \, Q_{u,v} \quad \text{and} \quad F_{u,t} =: \sum_{v \in \mathcal{V}} \mathbb{1}(\mathrm{cnt}_{u,v} = t) \, \hat{Q}_{u,v}^{\mathrm{gt}} \tag{14}$$

As a starting point, we will first derive our results with the genie-aided estimator, where we have oracle access to the ration of expected frequencies for every user, and then switch the analysis to the more general version when we do not. The genie-aided local estimator $\hat{Q}_{u,v}^{\mathrm{gen}}$ is:

$$\hat{Q}_{u,v}^{\mathrm{gen}} \propto \frac{\mathrm{cnt}_{u,v} + 1}{m} \cdot \frac{\mathbb{E}[\phi_{u,\mathrm{cnt}_{u,v}+1}]}{\mathbb{E}[\phi_{u,\mathrm{cnt}_{u,v}}]}, \tag{15}$$

and the corresponding average of genie-aided Good-Turing estimates:

$$\tilde{P}_v^{\mathrm{gen}} = \frac{1}{n} \sum_u \hat{Q}_{u,v}^{\mathrm{gen}}.$$

Next, we derive results for the setting where $m_u \sim \mathrm{Poi}(m)$ so that the local frequency counts of words become independent. Then, using the results on Poisson sampling in Mitzenmacher and Upfal [60], we can convert the bounds to the setting where each user samples exactly $m$ tokens.

We will use $\mathbb{1}_v^t$ as a shorthand for $\mathbb{1}(\lfloor \overline{\mathrm{cnt}}_v \rfloor = t)$ and similarly use $\mathbb{1}_{u,v}^t$ as a shorthand for $\mathbb{1}(\mathrm{cnt}_{u,v} = t)$. The following quantities define the global and local frequencies of average and local counts respectively:

$$\phi_t =: \sum_{v \in \mathcal{V}} \mathbb{1}_v^t \quad \text{and} \quad \phi_{u,t} = \sum_{v \in \mathcal{V}} \mathbb{1}_{u,v}^t. \tag{16}$$

For $F_t$ defined using the genie-aided estimator, to bound the competetive KL estimation error we first need to bound the KL distance between ground truth $M_t$ and estimate $F_t$ under high probability over the draw of the meta-dataset $\mathcal{S}$.

$$\mathrm{KL}(M_t \| F_t) = \sum_t M_t \log \frac{M_t}{F_t} \leq \log \left( \sum_t \frac{M_t^2}{F_t} \right)$$

$$= \log \left( 1 + \sum_t \frac{M_t - F_t^2}{F_t} \right) \leq \sum_t \frac{(M_t - F_t)^2}{F_t} \tag{17}$$

Using Lemma E.6 we lower bound the denominator in the KL upper bound in Eq. 17.

**Lemma E.6** (Lower bound on $F_t$). *With probability atleast $1 - nm^{-2}$, $F_t = \tilde{\Omega}\left(\frac{t\phi_t}{m}\right)$.*

*Proof.* The proof relies on three results for local Good-Turing (per-user) estimates. We will combine the three results, and then apply a union bound over all the users. Let $v$ be some token in the vocabulary that has an average count of $t$, *i.e.*, $\mathbb{1}_v^t = 1$. From Lemma 15, 16 and Claim 20 in Acharya et al. [2], with probability at least $1 - m^{-3}$, $\widehat{Q}_u^{\mathrm{un}} = \tilde{\Omega}(\mathrm{cnt}_{u,v}/m)$, where $\widehat{Q}_u^{\mathrm{un}}$ is the unnormalized local Good-Turing estimate. Additionally, from Lemma 17 in Acharya et al. [2], the normalization factor $N_u$ for user $u$ satisfies with probability at least $1 - 10m^{-2}$, $N_u = 1 + \tilde{O}(1/m^{1/4})$. Since $m_u$ is a Poisson distribution from Lemma D.1, the number of data points $m_u$ for the user satisfies $|m_u - m| \leq \sqrt{m}$, with constant probability. Putting these together, and applying a union bound over all $n$ users we conclude:

$$F_t = \tilde{\Omega}\left( \frac{1}{n} \sum_{v \in \mathcal{V}} \sum_u \frac{\mathbb{1}_v^t \mathrm{cnt}_{u,v}}{m} \right) = \tilde{\Omega}\left( \frac{\overline{\mathrm{cnt}}_v \phi_t}{m} \right) = \tilde{\Omega}\left( \frac{t\phi_t}{m} \right),$$

with probability at least $1 - nm^{-2}$. $\qquad\square$

Next we try to upper bound $(M_t - F_t)^2$ for some $t \geq 1$. But before that we introduce some more helpful results.

**Lemma E.7** (User's $Q_u$ concentrates around center $P$). *With probability at least $1 - \delta$, $\forall v \in \mathcal{V}$:*

$$\left| \frac{1}{n} \sum_u Q_{u,v} - P_v \right| = \mathcal{O}\left( \sqrt{\frac{P_v \log d/\delta}{n(1+\alpha)}} \right)$$

*Proof.* Since $Q_{u,v} \in [0,1]$ is a bounded random variable with variance $P_v(1-P_v)/(1+\alpha)$, we can apply the Bernstein inequality in Lemma D.2. This would give us a confidence interval for a single token $v$, which we can then union bound over all tokens $v \in \mathcal{V}$ to get the result. $\square$

**Lemma E.8** (variance of linear estimators; Claim 21 in Acharya et al. [2]). *For every distribution $p$,*

$$Var\left( \sum_v \sum_t \mathbb{1}_v^t f(x,t) \right) \leq \sum_x \sum_t \mathbb{E}[\mathbb{1}_v^t] f^2(x,t).$$

*Proof.* By Poisson sampling, the multiplicities are independent. Furthermore, the variance of the sum of independent random variables is the sum of their variances. Hence,

$$\begin{aligned}
\mathrm{Var}\left( \sum_v \sum_t \mathbb{1}_v^t f(x,t) \right) &= \sum_v \mathrm{Var}\left( \sum_t \mathbb{1}_v^t f(x,t) \right) \\
&\leq \sum_v \mathbb{E}\left[ \left( \sum_t \mathbb{1}_v^t f(x,t) \right)^2 \right] \\
&= \sum_v \mathbb{E}\left[ \sum_t (\mathbb{1}_v^t f(x,t))^2 \right] \quad\quad\quad \text{(a)} \\
&= \sum_v \sum_t \mathbb{E}[\mathbb{1}_v^t] f^2(x,t). \quad\quad\quad \text{(b)}
\end{aligned}$$

For $t \neq t'$, $\mathbb{E}[\mathbb{1}_v^t \mathbb{1}_v^{t'}] = 0$ and hence (a). (b) uses the fact that $\mathbb{1}_v^t$ is an indicator random variable. $\square$

**Lemma E.9** (expected sensitivity of the local frequencies; Claim 20 in [2]). *For every user distribution $p$ over $\mathcal{V}$, from which we draw $n$ i.i.d. samples, let $t$ be the local count and $\phi_t = \sum_{v \in \mathcal{V}} \mathbb{1}_v^t$ is the frequency of the local count,*

$$\mathbb{E}[\phi_t] - \mathbb{E}[\phi_{t+1}] = \mathcal{O}\left( \mathbb{E}[\phi_t] \max\left( \frac{\log m}{t+1}, \sqrt{\frac{\log n}{t+1}} \right) \right) + \frac{1}{n}.$$

*Proof.* We consider the two cases $t + 1 \geq \log n$ and $t + 1 < \log n$ separately. Consider the case when $t + 1 \geq \log n$. We first show that

$$\left| \mathbb{E}[1_x^t] - \mathbb{E}[1_x^{t+1}] \right| = e^{-np_x} \frac{(np_x)^t}{t!} \left| 1 - \frac{np_x}{t+1} \right| \leq 5 e^{-np_x} \frac{(np_x)^t}{t!} \sqrt{\frac{\log n}{t+1}} + \frac{2}{n^3}. \quad (4)$$

The first equality follows by substituting $\mathbb{E}[1_x^t] = e^{-np_x}(np_x)^t/t!$. For the inequality, note that if $|np_x - t - 1| \leq 25(t+1)\log n$, then the inequality follows. If not, then by the Chernoff bound $\mathbb{E}[1_x^t] = \Pr(t_x = t) \leq n^{-3}$ and hence $\left| \mathbb{E}[1_x^t] - \mathbb{E}[1_x^{t+1}] \right| \leq \mathbb{E}[1_x^t] + \mathbb{E}[1_x^{t+1}] \leq 2/n^3$.

By definition, $\mathbb{E}[\Phi_t] - \mathbb{E}[\Phi_{t+1}] = \sum_x \mathbb{E}[1_x^t] - \mathbb{E}[1_x^{t+1}]$. Substituting,

$$\begin{aligned}
|\mathbb{E}[\Phi_t] - \mathbb{E}[\Phi_{t+1}]| &\leq \sum_x \left| \mathbb{E}[1_x^t] - \mathbb{E}[1_x^{t+1}] \right| \\
&\overset{(a)}{=} \sum_x e^{-np_x} \frac{(np_x)^t}{t!} \left| 1 - \frac{np_x}{t+1} \right|
\end{aligned}$$

$$= \sum_{x:np_x \leq 1} e^{-np_x} \frac{(np_x)^t}{t!} \left| 1 - \frac{np_x}{t+1} \right| + \sum_{x:np_x > 1} e^{-np_x} \frac{(np_x)^t}{t!} \left| 1 - \frac{np_x}{t+1} \right|$$

$$\overset{(b)}{\leq} \sum_{x:np_x \leq 1} \frac{np_x}{t!} + \sum_{x:np_x > 1} 5 e^{-np_x} \frac{(np_x)^t}{t!} \sqrt{\frac{\log n}{t+1}} + \frac{2}{n^3}$$

$$\leq \frac{1}{n^2} + \mathcal{O}\left( \mathbb{E}[\Phi_t] \sqrt{\frac{\log n}{t+1}} \right) + \frac{2n}{n^3} \leq \mathcal{O}\left( \mathbb{E}[\Phi_t] \sqrt{\frac{\log n}{t+1}} \right) + \frac{1}{n}.$$

where (a) follows from the fact that $\mathbb{E}[1_x^t] = e^{-np_x}(np_x)^t/t!$. (b) follows from the fact that $np_x \leq 1$ in the first summation and Equation (4). The proof for the case $t + 1 < \log n$ is similar and hence omitted. □

**Lemma E.10** (Upper bound on $|M_t - F_t|$). *For any $t \geq 1$, with high probability at least $1 - n\mathrm{poly}(m)$.*

$$|M_t - F_t| = \tilde{\mathcal{O}}\left( \sqrt{t\mathbb{E}[\phi_t]} \cdot \left( \frac{1}{1+\alpha} + \frac{1}{m} \right) \right)$$

*Proof.*

$$|M_t - F_t| \leq \underbrace{\left| \frac{1}{n} \sum_{v \in \mathcal{V}} \sum_u 1_v^t (P_v - Q_{u,v}) \right|}_{A=:\text{Dirichlet concentration}} + \underbrace{\left| \frac{1}{n} \sum_{v \in \mathcal{V}} \sum_u 1_v^t (Q_{u,v} - \hat{Q}_{u,v}^{\mathrm{gt}}) \right|}_{B=:\text{Local Good-Turing}} \tag{18}$$

Our proof mainly relies on the concentration properties of Dirichlet distribution and some properties of the local Good-Turing estimate. While the first term is fairly streaightforward to bound, the second term requires us to handle the fact that $1_v^t$ is a random variable that depends on the global statistics for the token $v$, while $\hat{Q}_{u,v}^{\mathrm{gt}} - Q_{u,v}$ depends only on the local statistics. This requires us to analyze this differently from Acharya et al. [2].

Using Lemma E.8, followed by Lemma E.7 we can bound term $A$ in Eq. 18 in the following way:

$$\mathrm{Var}\left( \sum_{v \in \mathcal{V}} 1_v^t (P_v - \bar{cn}t_v/m) \right) \leq \sum_v \mathbb{E}[1_v^t] (P_v - \bar{cn}t_v/m)^2 \tag{19}$$

$$= \mathcal{O}\left( \mathbb{E}\left[ \sum_{v \in \mathcal{V}} \frac{1_v^t P_v}{(1+\alpha)} \right] \right) = \mathcal{O}\left( \mathbb{E}\left[ \sum_{v \in \mathcal{V}} \frac{1_v^t t}{m(1+\alpha)} \right] \right) \tag{20}$$

$$= \mathcal{O}\left( \frac{\mathbb{E}[\phi_t] t}{m(1+\alpha)} \right) \tag{21}$$

This completes our derivation for the upper bound on the first term ($A$) in Eq. 18. For the second term ($B$), we use the definition of the genie-aided local Good-Turing estimator and bound its variance in the following way:

$$\mathrm{Var}\left( \frac{1}{n} \sum_{v \in \mathcal{V}} \sum_u 1_v^t (Q_{u,v} - \hat{Q}_{u,v}^{\mathrm{gt}}) \right) \leq \sum_{v \in \mathcal{V}} \mathbb{E}[1_v^t] \left( \frac{1}{n} \sum_u Q_{u,v} - \hat{Q}_{u,v}^{\mathrm{gt}} \right)^2$$

$$= \sum_{v \in \mathcal{V}} \mathbb{E}[1_v^t] \left( \frac{1}{n} \sum_u Q_{u,v} - \frac{\mathrm{cnt}_{u,v}+1}{m_u} \frac{\mathbb{E}[\phi_{u,\mathrm{cnt}_{u,v}+1}]}{\mathbb{E}[\phi_{u,\mathrm{cnt}_{u,v}}]} \right)^2$$

$$\leq 2 \sum_{v \in \mathcal{V}} \mathbb{E}[1_v^t] \left( \frac{1}{n} \sum_u Q_{u,v} - \frac{\mathrm{cnt}_{u,v}+1}{m_u} \right)^2 \tag{22}$$

$$+ 2 \sum_{v \in \mathcal{V}} \mathbb{E}[1_v^t] \left( \frac{1}{n} \sum_u \frac{(\mathrm{cnt}_{u,v}+1)(\mathbb{E}[\phi_{u,\mathrm{cnt}_{u,v}+1} - \phi_{u,\mathrm{cnt}_{u,v}}])}{m_u \mathbb{E}[\phi_{u,\mathrm{cnt}_{u,v}}]} \right)^2$$

The first inequality above uses Lemma E.8, and the second equality uses the definition of the genie-aided estimator. In Eq. 22 we have two terms, and we bound the first term using Jensen inequality followed by the DKW inequality [70] for empirical estimators of the cumulative distribution. With high probability $1 - \delta$,

$$\sum_u \sum_{v \in \mathcal{V}} \mathbb{E}[\mathbb{1}_v^t] \left( \frac{1}{n} \sum_u Q_{u,v} - \frac{\text{cnt}_{u,v} + 1}{m_u} \right)^2 \leq \frac{1}{n} \sum_u \sum_{v \in \mathcal{V}} \left( Q_{u,v} - \frac{\text{cnt}_{u,v} + 1}{m_u} \right)^2$$

$$= \frac{1}{n} \sum_u \sum_{v \in \mathcal{V}} \mathbb{E}[\mathbb{1}_v^t] \mathcal{O} \left( \frac{\text{cnt}_{u,v} \log^2(d/\delta)}{m_u^2} \right)$$

$$\leq \sum_{v \in \mathcal{V}} \mathbb{E}[\mathbb{1}_v^t] \mathcal{O} \left( \frac{\overline{\text{cnt}_v} \log^2(d/\delta)}{m_u^2} \right)$$

$$\leq \tilde{\mathcal{O}} \left( \frac{t \mathbb{E}[\phi_t]}{m^2} \right)$$

The final inequality above uses the definition of $t$, and the concentration of $m_u$, *i.e.*, with high probability: $|m_u - \mathbb{E}[m_u]| = |m_u - m| = \tilde{\mathcal{O}}(\sqrt{m})$.

Next we will bound the second term in Eq. 22, for which we use some properties of the expected sensitivity of the local frequency counts (Lemma E.9).

$$\sum_{v \in \mathcal{V}} \mathbb{E}[\mathbb{1}_v^t] \left( \frac{1}{n} \sum_u \frac{(\text{cnt}_{u,v} + 1)(\mathbb{E}[\phi_{u,\text{cnt}_{u,v}+1} - \phi_{u,\text{cnt}_{u,v}}])}{m_u \mathbb{E}[\phi_{u,\text{cnt}_{u,v}}]} \right)^2$$

$$\leq \sum_{v \in \mathcal{V}} \mathbb{E}[\mathbb{1}_v^t] \left( \frac{1}{n} \sum_u \frac{(\text{cnt}_{u,v} + 1)}{m_u \sqrt{\text{cnt}_{u,v}}} \right)^2$$

Applying Jensen's inequality it is easu to see that we can bound the above term with:

$$\sum_{v \in \mathcal{V}} \mathbb{E}[\mathbb{1}_v^t] \left( \frac{1}{n} \sum_u \frac{(\text{cnt}_{u,v} + 1)(\mathbb{E}[\phi_{u,\text{cnt}_{u,v}+1} - \phi_{u,\text{cnt}_{u,v}}])}{m_u \mathbb{E}[\phi_{u,\text{cnt}_{u,v}}]} \right)^2 = \tilde{O} \left( \frac{t \mathbb{E}[\phi_t]}{m^2} \right) \tag{23}$$

Combining the result in Equation 19 with Equation 23, and then applying Lemma 8 from Orlitsky and Suresh [63] completes the proof of the Lemma. $\square$

Let us first consider the following simplified setting which deviates from our setup in Section 4 in two ways: (1) $Q_u = P$, *i.e.*, each user's true distribution matches exactly the cluster center (this matches the $\alpha \to \infty$ case in our setup); and (2) each user independently samples $m_u \sim \text{Poi}(m)$, which is the number of samples in their local dataset, that is used to estimate the empirical distribution $\widehat{Q}_u$ (note that $m_u = m$ for our results in Section 5). We shall now derive results in this setting, and then consider the more general case.

Without loss of generality, let us fix the center to be $c$, with some collection of users $\mathcal{Z}_c$. Then, we know that $\text{cnt}_v$ is distributed as a Poisson random variable with mean $m \sum_{u \in \mathcal{Z}_c} Q_u[v]$. Thus, with probability at least $1 - \delta$:

$$\left| \text{cnt}_u / n - \sum_{u \in \mathcal{Z}_c} m Q_u[v] / n \right| \leq \log(2/\delta).$$

We mainly rely on Lemma 13 for the Genie aided estimator from [2]. The key difference being that $\mathbb{E}[\phi_k]$, which is words with total count $k$ would differ from $\mathbb{E}[\phi_{\text{cnt}_{u,v}}]$. For, this we first bound $\lfloor k/n \rfloor - \text{cnt}_{u,v}$ with high probability. We do this, by:

$$\text{cnt}_{uv} - \mathbb{E}(\text{cnt}_{u,v}) =_\delta \tilde{O}(\sqrt{\mathbb{E}[\text{cnt}_{u,v}]})$$

$$=_\delta \tilde{O}(\sqrt{m Q_u[v]}) + m \tilde{O}\left(\sqrt{\frac{P[v]}{\alpha + 1}}\right)$$

since the counts follow Poisson distribution under indpendent sampling. Thus, when $\alpha = \Omega(m)$, the first term dominates, and we see that each this gives us:

$$\frac{k}{n} \in \left[ \mathrm{cnt}_{u,v} - \sqrt{\mathrm{cnt}_{u,v}}, \mathrm{cnt}_{u,v} + \sqrt{\mathrm{cnt}_{u,v}} \right]$$

with high probability. Then from Claim 20 in [2], qwe know that $|\mathbb{E}[\phi_k] - \mathbb{E}[\phi_{\mathrm{cnt}_{u,v}}]| = \mathcal{O}(k)$

The rest of our analysis relies with the genie-aided estimator mainly relies on Theorem 2 in Acharya et al. [2]. Using the result in Lemma E.10, we conclude that with probability $1 - n\mathrm{poly}(m)$ over the draw of the meta-dataset $\mathcal{S}$, the KL distance:

$$\mathrm{KL}(M_t \| F_t) \leq \tilde{O}\left( \sqrt{\frac{1}{m}} \cdot \frac{\alpha + m}{\alpha + 1} \right). \tag{24}$$

Next, we use Lemma 15 from Acharya et al. [2] to prove the same claim for the estimator that is not genie-aided. And finally, we reuse the conversion of the high probability result to one in expectation using arguments similar to our proof of Lemma 4.1. This completes our analysis of the upper bound on competetive regret for the average of local Good-Turings (our estimator).

Finally, for the lower bound in Theorem 4.3, we invoke known lower bounds on the concentration of the Dirichlet distribution $\tilde{\Omega}(1/\alpha+1)$, and minimax statistical lower bounds on the risk of the empirical estimator $\tilde{\Omega}(\frac{d}{m})$, to conclude that the competetive risk of FedAvg is at least $\tilde{\Omega}\left( d/m + 1/\alpha+1 \right)$. This completes our proof of Theorem 4.3. $\qquad\square$

**Lemma E.11** (Stirling approximation [46])**.** *The $\Gamma$ function is sandwiched as follows. There is a positive constant $C$, such that for all $x > 0$:*

$$Cx^{x-1/2}e^{-x} \ \leq \ \Gamma(x) \ \leq \ Cx^{x-1/2}e^{-x}e^{1/12x}.$$

*This implies $\log \Gamma(x) = \log C + (x - 1/2) \log x - x + O(1/x)$. For large enough $x$, we substitute $\Gamma(x) \approx \log C + (x - 1/2) \log x - x$.*

**Theorem E.12** (Clustering approximates MLE objective; restated)**.** *The joint likelihood for $P_c$ and $z_u$ can be upperbounded with the clustering objective in Eq. 4, and this approximation is tight upto an additive term of $O(1/\alpha)$.*

Let $\mathcal{V}$ be a fixed set of words, $d =: |\mathcal{V}|$ is the size of the set, and $\Delta(\mathcal{V})$ is the set of all possible discrete probability distributions over $\mathcal{V}$. Each user $u$ in the federated setup has an unknown distribution $Q_u \in \Delta(\mathcal{V})$. Further, each $Q_u$ belongs to one of $k$ clusters, and the cluster membership is denoted by $z_u$, where $z_u \in \mathbb{R}^k$ is a 1-hot $k$-dimensional vector. Each cluster $c \in [k]$ is associated with a center $P_c \in \Delta(\mathcal{V})$, and the matrix with column vectors as cluster centers is denoted as $\mathbf{P} \in \mathbb{R}^{d \times K}$, i.e., $\mathbf{P} =: [P_1, P_2, \ldots, P_k]$.

Independently for each user $u$, given the cluster membership $z_u$, and some $\alpha > 0$, the user's distribution $Q_u$ is sampled from a Dirichlet around the cluster center $\mathbf{P}z_u$, i.e., $Q_u \sim \mathrm{Dir}(\alpha \mathbf{P}z_u)$. Further, each user samples an i.i.d. dataset $\mathcal{S}_u$ of size $n_u$. When all users sample dataset of fixed size $n_u = n$, we refer to this setting as size-homogeneous, and the more general case as size-heterogeneous. Given the set of datasets $\{\mathcal{S}_u\}_u$ and the distribution assumptions above, we can write down the maximum-likelihood estimate (MLE) for the cluster centers $\mathbf{P}$ and cluster memberships $\{z_u\}_u$:

$$\arg\max_{\mathbf{P}, \{z_u\}_u} \quad \sum_u \log \Pr(\mathcal{S}_u \mid \mathbf{P}, \{z_u\}_u) \tag{25}$$

$$= \arg\max_{\mathbf{P}, \{z_u\}_u} \quad \sum_u \log \mathrm{DirMul}(n_u \widehat{Q}_u \mid \mathbf{P}, \{z_u\}_u)$$

$$= \arg\max_{\mathbf{P}, \{z_u\}_u} \quad \sum_u \log \Gamma(\alpha) + \log \Gamma(n_u + 1) - \log \Gamma(n_u + \alpha)$$

$$+ \left( \sum_x \log \Gamma(\widehat{Q}_u[x] + \alpha \mathbf{P}z_u[x]) - \log \Gamma(\alpha \mathbf{P}z_u[x]) - \log \Gamma(\widehat{Q}_u[x] + 1) \right),$$

where $\widehat{Q}_u$ is the empirical distribution obtained from the dataset $\mathcal{S}_u$ and $\gamma(x) =: \int t^{x-1} e^{-x} \, \mathrm{d}x$ is the Gamma function. In the above derivation, the second inequality follows from $n_u \widehat{Q}_u$ being a sufficient statistic for the class of distributions parameterized by $\{Q_u\}_u$.

For the size-homogenous case, we can simplify the MLE objective further to be:

$$\arg\max_{\mathbf{P},\{z_u\}_u} \quad \sum_u \sum_x \log \Gamma(\widehat{Q}_u[x] + \alpha \mathbf{P} z_u[x]) - \log \Gamma(\alpha \mathbf{P} z_u[x]) \tag{26}$$

For large enough $\alpha$, we can use the Stirling's approximation of the Gamma function to approximate the above MLE objective (see Lemma E.11):

$$\arg\max_{\mathbf{P},\{z_u\}_u} \quad \sum_u \sum_x (\widehat{Q}_u[x] + \alpha \mathbf{P} z_u[x] - 1/2) \log(\widehat{Q}_u[x] + \alpha \mathbf{P} z_u[x]) - (\widehat{Q}_u[x] + \alpha \mathbf{P} z_u[x])$$
$$- (\alpha \mathbf{P} z_u[x] - 1/2) \log(\alpha \mathbf{P} z_u[x]) + \alpha \mathbf{P} z_u[x] \tag{27}$$

$$= \arg\max_{\mathbf{P},\{z_u\}_u} \quad \sum_u \sum_x \widehat{Q}_u[x] \log(\widehat{Q}_u[x] + \alpha \mathbf{P} z_u[x]) + (\alpha \mathbf{P} z_u[x] - 1/2) \log\left(\frac{\widehat{Q}_u[x] + \alpha \mathbf{P} z_u[x]}{\alpha \mathbf{P} z_u x}\right)$$

$$= \arg\max_{\mathbf{P},\{z_u\}_u} \quad \sum_u \sum_x \widehat{Q}_u[x] \log \mathbf{P} z_u[x] + (\widehat{Q}_u[x] + \alpha \mathbf{P} z_u[x] - 1/2) \log\left(1 + \frac{\widehat{Q}_u[x]}{\alpha \mathbf{P} z_u[x]}\right)$$

$$= \arg\max_{\mathbf{P},\{z_u\}_u} \quad \sum_u -\text{KL}(\widehat{Q}_u \parallel \mathbf{P} z_u) + \sum_x (\widehat{Q}_u[x] + \alpha \mathbf{P} z_u[x] - 1/2) \log\left(1 + \frac{\widehat{Q}_u[x]}{\alpha \mathbf{P} z_u[x]}\right)$$

$$= \arg\min_{\mathbf{P},\{z_u\}_u} \quad \sum_u \text{KL}(\widehat{Q}_u \parallel \mathbf{P} z_u) + \sum_x 1/2 \log\left(1 + \frac{\widehat{Q}_u[x]}{\alpha \mathbf{P} z_u[x]}\right)$$
$$- \sum_x (\widehat{Q}_u[x] + \alpha \mathbf{P} z_u[x]) \log\left(1 + \frac{\widehat{Q}_u[x]}{\alpha \mathbf{P} z_u[x]}\right)$$

$$\leq \arg\min_{\mathbf{P},\{z_u\}_u} \quad \sum_u \text{KL}(\widehat{Q}_u \parallel \mathbf{P} z_u) + \sum_x 1/2 \frac{\widehat{Q}_u[x]}{\alpha \mathbf{P} z_u[x]}$$

$$= \arg\min_{\mathbf{P},\{z_u\}_u} \quad \sum_u \text{KL}(\widehat{Q}_u \parallel \mathbf{P} z_u) + \mathcal{O}\left(\frac{1}{\alpha}\right),$$

when cluster center $P[v] = \Omega(1)$.

## F   Missing proofs from Section 3

**Lemma F.1** ($\widehat{Q}_u$ variance). *For user $u$ from cluster $c$, the variance of the estimate $\widehat{Q}_u[v]$ is:*

$$\frac{P_c[v]}{\alpha + 1}\left(1 - \frac{1}{m_u}\right) + \frac{P_c[v](1 - P_c[v])}{m_u}.$$

*Proof.* By the Law of Total Variation, the variance of $\widehat{Q}_u[v]$ is:

$$\text{Var}(\widehat{Q}_u[v]) = \mathbb{E}_{Q_u[v]}[\text{Var}_{Q_u[v]}(\widehat{Q}_u[v] \mid Q_u[v])] + \text{Var}_{Q_u[v]}(\mathbb{E}[\widehat{Q}_u[v] | Q_u[v]])$$
$$= \mathbb{E}[\frac{1}{m_u} Q_u[v](1 - Q_u[v])] + \text{Var}(Q_u[v])$$
$$= \frac{1}{m_u}(P_c[v] - \frac{1}{\alpha + 1} - P_c[v]^2) + \frac{P_c[v]}{\alpha + 1}$$
$$= \frac{1}{m_u}(P_c[v](1 - P_c[v])) + (1 - \frac{1}{m_u})\frac{P_c[v]}{\alpha + 1}.$$

$\square$

## F.1 Proof of Theorem 3.2

**Theorem F.2** (Alg. 3 convergence; restated). *For the model in Section 4, if $\forall c, v$, $n_c = \Theta(K^2), P_c[v] = \Omega(1)$, centers are sufficiently separated, i.e., $\Delta =: \min_{i \neq j} \mathrm{KL}(P_i \| P_j) = \Omega(k^2 + k^3 d/n)$, and $\lambda =: \max_{i \neq j} \mathrm{KL}(P_i \| P_j)/\Delta$. Given an initialization with assignment error $O(\sqrt{1/\lambda})$ for any cluster, after $t = \log(|\mathcal{U}|)$ iterations, w.h.p. the assignment error is $O(\exp(-\Delta(\alpha + 1)))$.*

**Lemma F.3** (Pinsker [68]). *For $P, Q \in \Delta^{|\mathcal{V}|-1}$, the $\mathrm{KL}(P \| Q) \geq \frac{1}{2}\|P - Q\|_1^2 = 2\mathrm{TV}^2(P, Q)$.*

**Lemma F.4** (KL upper bound). *For $P, Q \in \Delta^{|\mathcal{V}|-1}$, the KL divergence $\mathrm{KL}(P \| Q) \leq \chi^2(P \, Q)$*

**Proof Overview.** The main technique we follow is to use Pinsker's inequality to lower bound KL divergence with $\mathrm{TV}^2(P, Q)$ and upper bound KL with $\chi^2$. This allows us to then treat the clustering in the $l_2^2$ metric, and pay the $d$-dimensional penalty term $\frac{1}{d}$. Based on this general principle, the following section presents the EM convergence analysis of Algorithm 3, extending results from [7] to our setting. While the steps below closely mirrors their analysis upto the final terms where we need to upper and lower bound KL, we present the full proof for completeness.

Let us begin by introducing some notaion. For any $S \subseteq [n]$, define $W_S = \sum_{i \in S} w_i$. Recall that $T_g^* = \{i \in [n], z_i = g\}$ and $T_g^{(s)} = \left\{i \in [n], \hat{z}_i^{(s)} = g\right\}$, let us define

$$S_{gh}^{(s)} = \left\{i \in [n], z_i = g, \hat{z}_i^{(s)} = h\right\} = T_g^* \cap T_h^{(s)}.$$

Then we have $n_h^{(s)} = \sum_{g \in [k]} n_{gh}^{(s)}$ and $n_h^* = \sum_{g \in [k]} n_{hg}^{(s)}$.

The mis-clustering rate at iteration $s$ can be written as

$$A_s = \frac{1}{n} \sum_{i=1}^{n} \mathbb{I}\{\hat{z}_i^{(s)} \neq z_i\} = \frac{1}{n} \sum_{g \neq h \in [k]^2} n_{gh}^{(s)}.$$

We define a cluster-wise mis-clustering rate at iteration $s$ as

$$G_s = \max_{h \in [k]} \left\{ \frac{\sum_{g \neq h \in [k]} n_{gh}^{(s)}}{n_h^{(s)}}, \frac{\sum_{g \neq h \in [k]} n_{hg}^{(s)}}{n_h^*} \right\}.$$

The first term in the maximum operator of definition of $G_s$ can be understood as the false positive rate of cluster $h$ and the second term is the true negative rate of cluster $h$. It is easy to see the relationship that $A_s \leq G_s$.

Let $\Delta = \min_{g \neq h \in [k]} \|\theta_g - \theta_h\|$ be the signal strength. For $h \in [k]$, let $\hat{\theta}_h^{(s)}$ be the estimated center of cluster $h$ at iteration $s$. Define our error rate of estimating centers at iteration $s$ as

$$\Lambda_s = \max_{h \in [k]} \frac{1}{\Delta} \|\hat{\theta}_h^{(s)} - \theta_h\|.$$

**Lemma F.5.** *Based on our definition of $M$ above, $G_0 = \mathcal{O}\left(\left(\frac{1}{2} - \frac{6}{\sqrt{r_k}}\right)\frac{1}{\lambda}\right)$.*

*Proof.* First we show $\lambda = O\sqrt{d}$. Since $P_c[v] = \frac{1}{d}$, we can invoke the KL upper bound above to show that $\frac{\Delta}{\lambda} = O(\frac{1}{d})$, and furthermore, the KL for each pair is $\Theta(M)$. Thus, this tells us:

$$\Delta(\alpha + 1) = \Omega(\sqrt{k^2 + \frac{k^3 d}{n}}). \tag{28}$$

In other words the signal to noise ratio is $\Delta/\sigma \geq \sqrt{k^2 + \frac{k^3 d}{n}}$. $\qquad \square$

Based on the above result, we also get:

$$\frac{\Delta}{\sigma}\sqrt{\frac{1/K}{1 + kd/n}} \geq C\sqrt{k}.$$

Recall $\sigma$ for us is given by $\sqrt{1/\alpha}$. Thus, we satisfy all conditions needed for Theorem 3.2 in their paper, the proof of which we replicate below for reader's convenience.

They assume the following initialization condition:

$$G_0 < \left(\frac{1}{2} - \frac{6}{\sqrt{r_k}}\right)\frac{1}{\lambda} \quad \text{or} \quad \Lambda_0 \le \frac{1}{2} - \frac{4}{\sqrt{r_k}}, \tag{29}$$

**Lemma F.6.**

$$\|W_S\| \le \sigma\sqrt{3(n+d)|S|} \quad \text{for all } S \subseteq [n]. \tag{30}$$

*with probability greater than* $1 - \exp(-0.3n)$.

**Lemma F.7.**

$$\lambda_{max}\left(\sum_{i=1}^n w_i w_i'\right) \le 6\sigma^2(n+d). \tag{31}$$

*with probability greater than* $1 - \exp(-0.5n)$.

**Lemma F.8.** *For any fixed* $i \in [n]$, $S \subseteq [n]$, $t > 0$ *and* $\delta > 0$,

$$\Pr\left\{\left\langle w_i, \frac{1}{|S|}\sum_{j\in S} w_j\right\rangle \ge \frac{3\sigma^2(t\sqrt{|S|}+d+\log(1/\delta))}{|S|}\right\} \le \exp\left(-\min\left\{\frac{t^2}{4d},\frac{t}{4}\right\}\right) + \delta.$$

**Lemma F.9.**

$$\|W_{T_h^*}\| \le 3\sigma\sqrt{(d+\log n)|T_h^*|} \quad \text{for all } h \in [k] \tag{32}$$

*with probability greater than* $1 - n^{-3}$.

**Lemma F.10.** *For any fixed* $\theta_1, \cdots, \theta_k \in \mathbb{R}^d$ *and* $a > 0$, *we have*

$$\sum_{i\in T_g^*} \mathbb{I}\left\{a\|\theta_h - \theta_g\|^2 \le \langle w_i, \|\theta_h - \theta_g\|\rangle\right\} \le n_g^* \exp\left(-\frac{a^2\Delta^2}{2\sigma^2}\right) + \sqrt{5n_g^* \log n} \tag{33}$$

*for all* $g \ne h \in [k]^2$ *with probability greater than* $1 - n^{-3}$.

The following two lemmas give the iterative relationship between the error of estimating centers and the error of estimating labels. Let $\mathcal{E}$ be the intersection of high probability events in Lemma F.6, Lemma F.7 Lemma F.9, Lemma F.10 and the initialization condition (29). Then we have $\mathbb{P}\{\mathcal{E}^c\} \le 3n^{-3} + \nu$. In the rest part of the proof, if not otherwise stated, we all condition on the event $\mathcal{E}$ and the following analysis are deterministic.

**Lemma F.11.** *On event* $\mathcal{E}$, *if* $G_s \le \frac{1}{2}$, *then we have*

$$\Lambda_s \le \frac{3}{r} + \min\left\{\frac{3}{r}\sqrt{kG_s} + 2G_s\Lambda_{s-1}, \lambda G_s\right\}. \tag{34}$$

**Lemma F.12.** *On event* $\mathcal{E}$, *if* $\Lambda_s \le \frac{1-\epsilon}{2}$ *and* $r \ge 36\epsilon^{-2}$, *then*

$$G_{s+1} \le \frac{2}{\epsilon^4 r^2} + \left(\frac{28}{\epsilon^2 r}\Lambda_s\right)^2 + \sqrt{\frac{5k\log n}{\alpha^2 n}}. \tag{35}$$

*Proof of Lemma F.11.* For any $B \subseteq [n]$, define $\bar{Y}_B = \frac{1}{|B|}\sum_{i\in B} y_i$. The error of estimated centers at step $s$ can be written as

$$\begin{aligned}
\hat{\theta}_h^{(s)} - \theta_h &= \frac{1}{n_h}\sum_{i\in S_{hh}}(y_i - \theta_h) + \frac{1}{n_h}\sum_{a\ne h}\sum_{i\in S_{ah}}(y_i - \theta_h) \\
&= \frac{1}{n_h}\sum_{i\in S_{hh}} w_i + \sum_{a\ne h}\frac{n_{ah}}{n_h}\left(\bar{Y}_{S_{ah}} - \theta_h\right)
\end{aligned}$$

According to our label update step, we have $\|y_i - \hat{\theta}_h^{(s-1)}\| \le \|y_i - \hat{\theta}_a^{(s-1)}\|$ for any $i \in S_{ah}$. This means for any $i \in S_{ah}$, $y_i$ is closer to $\hat{\theta}_h^{(s-1)}$ than $\hat{\theta}_a^{(s-1)}$, so is the average of $\{y_i, i \in S_{ah}\}$. Thus, we have

$$\|\bar{Y}_{S_{ah}} - \hat{\theta}_h^{(s-1)}\| \le \|\bar{Y}_{S_{ah}} - \hat{\theta}_a^{(s-1)}\|.$$

Consequently, triangle inequality gives us

$$\left\|\bar{Y}_{S_{ah}} - \theta_h\right\| \le \left\|\bar{Y}_{S_{ah}} - \theta_a\right\| + \|\hat{\theta}_a^{(s-1)} - \theta_a\| + \|\hat{\theta}_h^{(s-1)} - \theta_h\|,$$

which, combined with Lemma F.6 and the definition of $\Lambda_{s-1}$, yields

$$\left\|\bar{Y}_{S_{ah}} - \theta_h\right\| \le \sigma\sqrt{3(n+d)/n_{ah}} + 2\Lambda_{s-1}\Delta.$$

Taking a weighted sum over $a \ne h \in [k]$, we get

$$
\begin{aligned}
\sum_{a \ne h} \frac{n_{ah}}{n_h} \left\|\bar{Y}_{S_{ah}} - \theta_h\right\| &\le \frac{\sigma\sqrt{3(n+d)}}{n_h} \sum_{a \ne h} \sqrt{n_{ah}} + 2\Lambda_{s-1}\Delta \sum_{a \ne h} \frac{n_{ah}}{n_h} \\
&\le \frac{\sigma\sqrt{3(n+d)}}{\sqrt{n_h}} \sqrt{(k-1)G_s} + 2G_s\Lambda_{s-1}\Delta,
\end{aligned}
$$

where the Last inequality is due to Cauchy-Schwartz and the fact that $\sum_{a \ne h} n_{ah} \le G_s n_h$. Note that $W_{S_{hh}} = W_{T_h^*} - \sum_{a \ne h} W_{S_{ha}}$. Triangle inequality and Lemma F.10 imply

$$\|W_{S_{hh}}\| \le 3\sigma\sqrt{d + \log n}\sqrt{n_h^*} + \sigma\sqrt{3(n+d)}\sqrt{n_h^* - n_{hh}}.$$

Since $G_s \le \frac{1}{2}$, we have

$$n_h \ge n_{hh} \ge n_h^*(1 - G_s) \ge \frac{1}{2}n_h^* \ge \frac{1}{2}\alpha n. \tag{36}$$

Combining the pieces, we obtain

$$
\begin{aligned}
\left\|\hat{\theta}_h^{(s)} - \theta_h\right\| &\le 3\sigma\sqrt{\frac{d + \log n}{\alpha n}} + 3\sigma\sqrt{\frac{k(n+d)}{\alpha n}G_s} + 2G_s\Lambda_{s-1}\Delta \\
&\le \left(\frac{3}{r}(1 + \sqrt{kG_s}) + 2G_s\Lambda_{s-1}\right)\Delta. \tag{37}
\end{aligned}
$$

Therefore, we get the first term in (34). To prove the second term, we decompose $\hat{\theta}_h^{(s)}$ differently.

$$
\begin{aligned}
\hat{\theta}_h^{(s)} &= \frac{1}{n_h} \sum_{i=1}^{n} (\theta_{z_i} + w_i) \mathbb{I}\left\{\hat{z}_i^{(s)} = h\right\} \\
&= \frac{1}{n_h} \sum_{a=1}^{k} \sum_{i=1}^{n} \theta_a \mathbb{I}\left\{z_i = a, \hat{z}_i^{(s)} = h\right\} + \frac{1}{n_h} \sum_{i \in T_h} w_i \\
&= \sum_{a=1}^{k} \frac{n_{ah}}{n_h}\theta_a + \frac{1}{n_h}W_{T_h}. \tag{38}
\end{aligned}
$$

Then, the error of $\hat{\theta}_h^{(s)}$ can be upper bounded as

$$\left\|\hat{\theta}_h^{(s)} - \theta_h\right\| = \left\|\sum_{a=1}^{k} \frac{n_{ah}}{n_h}(\theta_a - \theta_h) + \frac{1}{n_h}W_{T_h}\right\| \le \left\|\sum_{a \ne h} \frac{n_{ah}}{n_h}(\theta_a - \theta_h)\right\| + \left\|\frac{1}{n_h}W_{T_h}\right\|.$$

By triangle inequality,

$$\left\|\sum_{a \ne h} \frac{n_{ah}}{n_h}(\theta_a - \theta_h)\right\| \le \sum_{a \ne h} \frac{n_{ah}}{n_h}\|\theta_a - \theta_h\| \le \lambda\Delta \sum_{a \ne h} \frac{n_{ah}}{n_h} \le \lambda\Delta G_s. \tag{39}$$

This, together with Lemma F.6 and (36), implies

$$\left\|\hat{\theta}_h^{(s)} - \theta_h\right\| \le \lambda\Delta G_s + \sigma\sqrt{\frac{3(n+d)}{n_h}} \le \left(\lambda G_s + \frac{3}{r}\right)\Delta \tag{40}$$

for all $h \in [k]$. The proof is complete.

$\square$

*Proof of Lemma F.12.* For any $g \neq h \in [k] \times [k]$,

$$\mathbb{I}\left\{z_i = g, \hat{z}_i^{(s+1)} = h\right\} \leq \mathbb{I}\left\{\|\theta_g + w_i - \hat{\theta}_h^{(s)}\|^2 \leq \|\theta_g + w_i - \hat{\theta}_g^{(s)}\|^2\right\}$$

$$= \mathbb{I}\left\{\|\theta_g - \hat{\theta}_h^{(s)}\|^2 - \|\theta_g - \hat{\theta}_g^{(s)}\|^2 \leq 2\left\langle w_i, \hat{\theta}_h^{(s)} - \hat{\theta}_g^{(s)}\right\rangle\right\}. \quad (41)$$

Triangle inequality implies

$$\|\theta_g - \hat{\theta}_h^{(s)}\|^2 \geq \left(\|\theta_g - \theta_h\| - \|\theta_h - \hat{\theta}_h^{(s)}\|\right)^2 \geq (1 - \Lambda_s)^2 \|\theta_g - \theta_h\|^2.$$

Using the fact that $(1-x)^2 - y^2 \geq (1 - x - y)^2$ when $y(1 - x - y) \geq 0$, we obtain

$$\|\theta_g - \hat{\theta}_h^{(s)}\|^2 - \|\theta_g - \hat{\theta}_g^{(s)}\|^2 = (1 - 2\Lambda_s)^2 \|\theta_g - \theta_h\|^2 \geq \epsilon^2 \|\theta_g - \theta_h\|^2. \quad (42)$$

Denote by $\Delta_h = \hat{\theta}_h^{(s)} - \theta_h$ for $h \in [k]$. Then,

$$\mathbb{I}\left\{z_i = g, \hat{z}_i^{(s+1)} = h\right\}$$

$$\leq \mathbb{I}\left\{\epsilon^2 \|\theta_g - \theta_h\|^2 \leq 2\left\langle w_i, \theta_h - \theta_g + \Delta_h - \Delta_g\right\rangle\right\}$$

$$\leq \mathbb{I}\left\{\frac{\epsilon^2}{2}\|\theta_g - \theta_h\|^2 \leq 2\left\langle w_i, \theta_h - \theta_g\right\rangle\right\} + \mathbb{I}\left\{\frac{\epsilon^2}{2}\Delta^2 \leq 2\left\langle w_i, \Delta_h - \Delta_g\right\rangle\right\}.$$

Taking a sum over $i \in T_g^*$ and using Markov's inequality on the second term, we obtain

$$n_{gh}^{(s+1)} \leq \sum_{i \in T_g^*} \mathbb{I}\left\{\frac{\epsilon^2}{4}\|\theta_g - \theta_h\|^2 \leq \left\langle w_i, \theta_h - \theta_g\right\rangle\right\} + \sum_{i \in T_g^*} \frac{16}{\epsilon^4 \Delta^4}\left(w_i'(\Delta_h - \Delta_g)\right)^2 \quad (43)$$

Note that $\mathbb{I}\left\{\frac{\epsilon^2}{4}\|\theta_g - \theta_h\|^2 \leq \left\langle w_i, \theta_h - \theta_g\right\rangle\right\}$ are independent Bernoulli random variables. By Lemma F.10, the first term in RHS of (43) can be upper bounded by

$$n_g^* \exp\left(-\frac{\epsilon^4 \Delta^2}{32\sigma^2}\right) + \sqrt{5 n_g^* \log n}. \quad (44)$$

By Lemma F.7, the second term in RHS of (43) can be upper bounded by

$$\sum_{i \in T_g^*} \frac{16}{\epsilon^4 \Delta^4}\left(w_i'(\Delta_h - \Delta_g)\right)^2 \leq \frac{96(n_g^* + d)\sigma^2}{\epsilon^4 \Delta^4}\|\Delta_g - \Delta_h\|^2. \quad (45)$$

Combining (43), (44) and (45) and using the fact that $\|\Delta_g - \Delta_h\|^2 \leq 4\Lambda_s^2 \Delta^2$, we get

$$n_{gh}^{(s+1)} \leq n_g^* \exp\left(-\frac{\epsilon^4 \Delta^2}{32\sigma^2}\right) + \sqrt{5 n_g^* \log n} + \frac{384(n_g^* + d)\sigma^2}{\epsilon^4 \Delta^2}\Lambda_s^2.$$

Consequently,

$$\max_{g \in [k]} \sum_{h \neq g} \frac{n_{gh}^{(s+1)}}{n_g^*} \leq k \exp\left(-\frac{\epsilon^4 \Delta^2}{32\sigma^2}\right) + k\sqrt{\frac{5 \log n}{\alpha n}} + \frac{384}{\epsilon^4 r^2}\Lambda_s^2. \quad (46)$$

Since $\Lambda_s \leq 1/2$ and $r \geq 20\epsilon^{-2}$, the RHS of (46) is smaller that $1/2$ when $\alpha n \geq 32 k^2 \log n$. Thus,

$$n_h^{(s+1)} \geq n_{hh}^{(s+1)} \geq \frac{1}{2} n_h^* \geq \frac{1}{2}\alpha n$$

for all $h \in [k]$ and we have

$$\max_{h \in [k]} \sum_{g \neq h} \frac{n_{gh}^{(s+1)}}{n_h^{(s+1)}} \leq \frac{2}{\alpha} \exp\left(-\frac{\epsilon^4 \Delta^2}{32\sigma^2}\right) + \sqrt{\frac{5k \log n}{\alpha^2 n}} + \frac{768}{\epsilon^4 r^2}\Lambda_s^2, \quad (47)$$

which, together with (46), implies

$$G_{s+1} \leq \exp\left(-\frac{\epsilon^4 \Delta^2}{32\sigma^2} + \log(2/\alpha)\right) + \sqrt{\frac{5k \log n}{\alpha^2 n}} + \frac{768}{\epsilon^4 r^2}\Lambda_s^2$$

Under the assumptions that $\epsilon^4 \alpha \Delta^2 / \sigma^2 \geq r^2 \epsilon^4 \geq 36$, we have the desired result (35). $\qquad \square$

*Proof.* From Lemma F.11, a necessary condition for $\Lambda_0 \leq \frac{1}{2} - \frac{4}{\sqrt{r}}$ is $G_0 \leq (\frac{1}{2} - \frac{6}{\sqrt{r}})\frac{1}{\lambda}$. Setting $\epsilon = \frac{7}{\sqrt{r}}$ in Lemma F.12, we have $G_1 \leq 0.35$. Plugging it into Lemma F.11 gives us $\Lambda_1 \leq 0.4$, under the assumption that $r \geq 16\sqrt{k}$. Then it can be easily proved by induction that $G_s \leq 0.35$ and $\Lambda_s \leq 0.4$ for all $s \geq 1$. Consequently, Lemma F.11 yields

$$\Lambda_s \leq \frac{3}{r} + \frac{3}{r}\sqrt{kG_s} + G_s \leq \frac{1}{2} + G_s$$

which, combined with (35), implies

$$G_{s+1} \leq \frac{C}{r^2} + \frac{C}{r^2}\left(\frac{1}{4} + 2G_s + G_s^2\right) + \sqrt{\frac{5k\log n}{\alpha^2 n}} \leq \frac{2C}{r^2} + \frac{3C}{r^2}G_s + \sqrt{\frac{5k\log n}{\alpha^2 n}}$$

for some constant $C$. Here we have chosen $\epsilon = 1/5$ in Lemma 35 to get the first inequality. $\square$

### F.1.1 Combining Analysis in $\ell_2$ Norm and Mapping it Back KL divergence.

*Proof.* From the proof of Lemma F.11, the error of estimating $\theta_h$ at iteration $s$ can be written as $\hat{\theta}_h^{(s)} - \theta_h = \frac{1}{n_h}W_{T_h^*} + u_h$, with

$$\|u_h\| \leq \left(\frac{3}{r}\sqrt{kG_s} + G_s\right)\Delta \leq \sqrt{G_s}\Delta \tag{48}$$

In addition, by Lemma F.11 and Lemma F.12, there is a constant $C_1$ such that

$$\Lambda_s \leq \frac{3}{r} + \sqrt{G_s} + 2G_s\Lambda_{s-1} \leq \frac{C_1}{r} + \frac{C_1}{r}\Lambda_{s-1} + 0.7\Lambda_{s-1} + \left(\frac{C_1 k\log n}{\alpha^2 n}\right)^{1/4}$$

for all $s \geq 1$. Therefore, when $r$ is large enough, we have

$$\Lambda \leq C_2 r^{-1} + C_2\left(\frac{k\log n}{\alpha^2 n}\right)^{1/4}$$

for all $s \geq \log n$. Then by (42), we have

$$\mathbb{I}\left\{z_i = g, \hat{z}_i^{(s+1)} = h\right\} \leq \mathbb{I}\left\{\beta_1\|\theta_g - \theta_h\|^2 \leq 2\langle w_i, \theta_h - \theta_g + \Delta_h - \Delta_g\rangle\right\}$$

where $(1 - 2\Lambda_s)^2 \geq \beta_1 := 1 - 4C_2 r^{-1} - 4C_2\left(\frac{k\log n}{\alpha^2 n}\right)^{1/4}$.

In order to prove that $A_s$ attains convergence rates, we first upper bound the expectation of $A_s$ and then derive the high probability bound using Markov's inequality. Similar to the two-mixture case, we need to upper bound the inner product $\langle w_i, \Delta_h - \Delta_g\rangle$ more carefully. Note that $\{T_h^*, h \in [k]\}$ are deterministic sets, we could use concentration equalities to upper bound $W_{T_h^*}$ and $u_h$ parts separately.

Let $v_h = \frac{1}{n_h}W_{T_h^*}$ for $h \in [k]$ and we decompose $\mathbb{I}\left\{z_i = g, \hat{z}_i^{(s+1)} = h\right\}$ into three terms.

$$\begin{aligned}
\mathbb{I}\left\{z_i = g, \hat{z}_i^{(s+1)} = h\right\} \leq \quad &\mathbb{I}\left\{\beta\|\theta_g - \theta_h\|^2 \leq 2\langle w_i, \theta_h - \theta_g\rangle\right\} \\
&+ \mathbb{I}\left\{\beta_2\Delta^2 \leq 2\langle w_i, u_h - u_g\rangle\right\} \\
&+ \mathbb{I}\left\{\beta_4\Delta^2 \leq 2\langle w_i, v_h - v_g\rangle\right\},
\end{aligned}$$

where $\beta_2$ and $\beta_4$ will be specified later and $\beta = \beta_1 - \beta_2 - \beta_4$. Taking a sum over $h \in [k]$ and $i \in [n]$, we obtain

$$\mathbb{E}A_{s+1} \leq \mathbb{E}J_1 + \mathbb{E}J_2 + \mathbb{E}J_3$$

with

$$J_1 = \sum_{h\in[k]}\frac{1}{n}\sum_{i=1}^n \mathbb{I}\left\{\beta\|\theta_{z_i} - \theta_h\|^2 \leq 2\langle w_i, \theta_h - \theta_{z_i}\rangle\right\} \tag{49}$$

$$J_2 = \sum_{h\in[k]}\frac{1}{n}\sum_{i=1}^n \mathbb{I}\left\{\beta_2\Delta^2 \leq 2\langle w_i, u_h - u_{z_i}\rangle\right\}. \tag{50}$$

$$J_3 = \sum_{h\in[k]}\frac{1}{n}\sum_{i=1}^n \mathbb{I}\left\{\beta_4\Delta^2 \leq 2\langle w_i, v_{z_i} - v_h\rangle\right\}. \tag{51}$$

Let us first consider the expectation of $J_1$. Using Chernoff's bound, we have

$$\Pr\left\{\beta\|\theta_g - \theta_h\|^2 \leq 2\langle w_i, \theta_h - \theta_g\rangle\right\} \leq \exp\left(-\frac{\beta^2\|\theta_h - \theta_g\|^2}{8\sigma^2}\right) \leq \exp\left(-\frac{\beta^2\Delta^2}{8\sigma^2}\right).$$

Thus,

$$\mathbb{E}J_1 \leq k\exp\left(-\frac{\beta^2\Delta^2}{8\sigma^2}\right) = \exp\left(-\frac{\gamma\Delta^2}{8\sigma^2}\right),$$

with $\gamma = \beta^2 - \frac{8\sigma^2\log k}{\Delta^2} \geq \beta^2 - 8/r^2$.

We use Markov Inequality to upper bound $J_2$. Markov's inequality and Lemma F.7 give us

$$
\begin{aligned}
\frac{1}{n}\sum_{i=1}^{n}\mathbb{I}\left\{\beta_2\Delta^2 \leq 2\langle w_i, u_h - u_{z_i}\rangle\right\} &\leq \frac{4}{n\beta_2^2\Delta^4}\sum_{g\in[k]}\sum_{i\in T_g^*}(w_i'(u_h - u_g))^2 \\
&\leq \frac{24\sigma^2}{n\beta_2^2\Delta^4}\sum_{g\in[k]}(n_g^* + d)\|u_h - u_g\|^2.
\end{aligned}
$$

(48) implies

$$J_2 \leq \frac{96\sigma^2 G_s}{n\beta_2^2\Delta^2}\sum_{h\in[k]}\sum_{g\in[k]}(n_g^* + d) \leq \frac{96\sigma^2 k(n+kd)}{\alpha n\beta_2^2\Delta^2}A_s = \frac{12\sqrt{k}}{r}A_s.$$

Here the second inequality is due to the fact that $G_s \leq A_s/\alpha$. And we choose $\beta_2 = \sqrt{8k/r}$ in the last equality.

Finally, we upper bound the expectation of $J_3$. Given $z_i = g$, we have

$$
\begin{aligned}
&\Pr\left\{\beta_4\Delta^2 \leq 2\langle w_i, v_g - v_h\rangle\right\} \\
\leq\ &\Pr\left\{\frac{\beta_4}{4}\Delta^2 \leq \langle w_i, v_g\rangle\right\} + \Pr\left\{-\frac{\beta_4}{4}\Delta^2 \geq \langle w_i, v_h\rangle\right\} \\
\leq\ &\Pr\left\{\frac{\beta_4}{8}\Delta^2 \leq \left\langle w_i, \frac{1}{n_g^*}W_{T_g^*}\right\rangle\right\} + \Pr\left\{-\frac{\beta_4}{8}\Delta^2 \geq \left\langle w_i, \frac{1}{n_h^*}W_{T_h^*}\right\rangle\right\}
\end{aligned}
$$

Choosing $t = \max\{\frac{\sqrt{d}\Delta}{\sigma}, \frac{\Delta^2}{\sigma^2}\}$, $\delta = \exp\left(-\frac{\Delta^2}{4\sigma^2}\right)$ in Lemma F.8, and

$$\beta_4 = \frac{64}{r} \geq \frac{8}{\Delta^2}\left(\frac{3\max\{\sqrt{d}\sigma\Delta, \Delta^2\}}{\sqrt{\alpha n}} + \frac{3\sigma^2 d + \Delta^2}{\alpha n}\right),$$

we obtain $\Pr\left\{\beta_4\Delta^2 \leq 2\langle w_i, v_g - v_h\rangle\right\} \leq 2\exp(-\Delta^2/(4\sigma^2))$, where we have used the assumption that $n_g^* \geq \alpha n$ and $\alpha n \geq 36r^2$. Thus,

$$\mathbb{E}J_3 \leq 2k\exp\left(-\frac{\Delta^2}{\sigma^2}\right),$$

Combining the pieces, we have

$$
\begin{aligned}
\mathbb{E}A_{s+1} &\leq\ \mathbb{E}[J_1] + \mathbb{E}[J_2\mathbb{I}\{\mathcal{E}\}] + \mathbb{E}[J_3] + \mathbb{P}\{\mathcal{E}^c\} \\
&\leq\ \exp\left(-\frac{\gamma\Delta^2}{8\sigma^2}\right) + \frac{12\sqrt{k}}{r}\mathbb{E}A_s + 2k\exp\left(-\frac{\Delta^2}{\sigma^2}\right),
\end{aligned}
$$

with $\gamma = (\beta_1 - \sqrt{8k/r} - 64/r)^2 - 8/r^2 = 1 - o(1)$. Here only prove the case that $r \to \infty$. For the finite case, all the $o(1)$ in the following proof can be substituted by a small constant.

$$\mathbb{E}A_s \leq \frac{1}{2^{s-\lceil\log r\rceil}} + 2\exp\left(-(1-\eta)\frac{\Delta^2}{8\sigma^2}\right) + \frac{2}{n^3} \leq 2\exp\left(-(1-\eta)\frac{\Delta^2}{8\sigma^2}\right) + \frac{3}{n^3}$$

when $s \geq 4 \log n$. By Markov's inequality, for any $t > 0$,

$$\Pr\{A_s \geq t\} \leq \frac{1}{t}\mathbb{E}A_s \leq \frac{2}{t}\exp\left(-(1-\eta)\frac{\Delta^2}{8\sigma^2}\right) + \frac{3}{n^3 t}. \tag{52}$$

If $(1-\eta)\frac{\Delta^2}{8\sigma^2} \leq 2\log n$, choose $t = \exp\left(-(1-\eta-\frac{8\sigma}{\Delta})\frac{\Delta^2}{8\sigma^2}\right)$ and we have

$$\Pr\left\{A_s \geq \exp\left(-(1-\eta-\frac{8\sigma}{\Delta})\frac{\Delta^2}{8\sigma^2}\right)\right\} \leq \frac{4}{n} + 2\exp\left(-\frac{\Delta}{\sigma}\right).$$

Otherwise, since $A_s$ only takes discrete values of $\{0, \frac{1}{n}, \cdots, 1\}$, choosing $t = \frac{1}{n}$ in (52) leads to

$$\Pr\{A_s > 0\} = \Pr\left\{A_s \geq \frac{1}{n}\right\} \leq 2n\exp(-2\log n) + \frac{3}{n^2} \leq \frac{4}{n}.$$

Now, from Lemma F.3 and Lemma F.4, for $P_i[v] = \Omega(1), \forall i$, we conclude that $\mathrm{KL}(P_i\|P_j) = \Theta(\|P_i - P_j\|_2)$. Applying the result from above we then get:

$$r_k \geq C\sqrt{k}$$
$$\implies \quad \frac{\Delta}{\sigma} \geq c\sqrt{k^2(1+{kd}/{n})}$$

Since $n_c = \Omega(k^2)$ we have $n = \Omega(k^3)$, which implies:

$$n\alpha^2 \geq Ck\log n,$$

which is the first condition in the EM analysis. The second condition requires the KL to be sandwiched by $\ell_2$ norm. But this means that we we can replace $\lambda$ with $\sqrt{\lambda}$ and $\Delta$ with $\sqrt{\Delta}$ in the result in Eq. 52, and complete the proof of Theorem 3.2. □

## F.2 Privacy Analysis from Section 5

### F.2.1 Technical Lemmas

We begin by recalling the definition of differential privacy, and the variant of concentrated differential privacy that we use in this work.

**Definition F.13** (Differential Privacy (DP) [24]). A randomized algorithm $M : \mathcal{X}^n \to \mathcal{Y}$ satisfies $(\epsilon, \delta)$-differential privacy ($(\epsilon, \delta)$-DP) if for every pair of neighboring datasets $X, X' \in \mathcal{X}^n$ (i.e., datasets that differ in exactly one entry),

$$\forall Y \subseteq \mathcal{Y} \quad \Pr M(X) \in Y \leq e^\epsilon \cdot \Pr M(X') \in Y + \delta.$$

When $\delta = 0$, we say that $M$ satisfies $\epsilon$-differential privacy or pure differential privacy.

**Definition F.14** (Concentrated Differential Privacy (zCDP) [15]). A randomized algorithm $M : \mathcal{X}^n \to \mathcal{Y}$ satisfies $\rho$-zCDP if for every pair of neighboring datasets $X, X' \in \mathcal{X}^n$,

$$\forall \alpha \in (1, \infty) \quad D_\alpha\left(M(X)\|M(X')\right) \leq \rho\alpha,$$

where $D_\alpha\left(M(X)\|M(X')\right)$ is the $\alpha$-Rényi divergence between $M(X)$ and $M(X')$.[2]

Note that zCDP and DP are on different scales, but otherwise can be ordered from most-to-least restrictive. Specifically, $(\epsilon, 0)$-DP implies $\frac{\epsilon^2}{2}$-zCDP, which implies roughly $(\epsilon\sqrt{2\log(1/\delta)}, \delta)$-DP for every $\delta > 0$ [15].

Both these definitions are closed under post-processing and can be composed with graceful degradation of the privacy parameters.

**Lemma F.15** (Post Processing [15]). *If* $M : \mathcal{X}^n \to \mathcal{Y}$ *is* $(\epsilon, \delta)$*-DP, and* $P : \mathcal{Y} \to \mathcal{Z}$ *is any randomized function, then the algorithm* $P \circ M$ *is* $(\epsilon, \delta)$*-DP. Similarly if* $M$ *is* $\rho$*-zCDP then the algorithm* $P \circ M$ *is* $\rho$*-zCDP.*

---

[2]Given two probability distributions $P, Q$ over $\Omega$, $D_\alpha(P\|Q) = \frac{1}{\alpha-1}\log\left(\sum_x P(x)^\alpha Q(x)^{1-\alpha}\right)$.

**Lemma F.16** (Composition of CDP [15]). *If $M$ is an adaptive composition of differentially private algorithms $M_1, \ldots, M_T$, then*

1. *if $M_1, \ldots, M_T$ are $(\epsilon_1, \delta_1), \ldots, (\epsilon_T, \delta_T)$-DP then $M$ is $(\sum_t \epsilon_t, \sum_t \delta_t)$-DP, and*

2. *if $M_1, \ldots, M_T$ are $\rho_1, \ldots, \rho_T$-zCDP then $M$ is $(\sum_t \rho_t)$-zCDP.*

We can achieve differential privacy via noise addition proportional to sensitivity [24].

**Definition F.17** (Sensitivity). Let $f : \mathcal{X}^n \to \mathbb{R}^d$ be a function, its $\ell_2$-*sensitivity* is defined to be $\Delta_{f,2} = \max_{X \sim X' \in \mathcal{X}^n} \|f(X) - f(X')\|_2$, Here, $X \sim X'$ denotes that $X$ and $X'$ are neighboring datasets (i.e., those that differ in exactly one entry).

For functions with bounded $\ell_1$-sensitivity, we can achieve $\epsilon$-DP by adding noise from a Laplace distribution proportional to $\ell_1$-sensitivity. For functions taking values in $\mathbb{R}^d$ for large $d$ it is more useful to add noise from a Gaussian distribution proportional to the $\ell_2$-sensitivity, to get $(\epsilon, \delta)$-DP and $\rho$-zCDP.

**Lemma F.18** (Gaussian Mechanism). *Let $f : \mathcal{X}^n \to \mathbb{R}^d$ be a function with $\ell_2$-sensitivity $\Delta_{f,2}$. Then the Gaussian mechanism*

$$M_f(X) = f(X) + N\left(0, \left(\frac{\Delta_{f,2}}{\sqrt{2\rho}}\right)^2 \cdot I_{d \times d}\right)$$

*satisfies $\rho$-zCDP.*

### F.2.2 Proof of Theorem 3.3

From the Lemmas above for Gaussian and Laplace mechanism, it is easy to see that the adaptive clipping step in Algorithm 4 satisfies $2\rho/3$-zCDP, where as the Laplace mechanism is $\rho/3$-zCDP. Then, we can use composition lemma to complete show $\rho$-zCDP of Algorithm 4.

For Algorithm 5, each step uses satisfies $\rho/K$-zCDP from the guarantees for exponential mechanism, since we clip the KL difference with $c$. Here too, composition gives us the result.

For the final end-to-end algorithm, we use the private aggregation scheme, we describe, and changing the user data would only affect two cluster estimates, but since we use add-one, leave-one definition of privacy, the entire step is $\rho/3T$-zCDP. Finally, composing over the $T$ rounds of clustering and with initial private center and private init we get the final result. We then use Lemma 3.6 from [15] to convert zCDP guarantees to $(\varepsilon, \delta)$-DP.

