# OpenReview forum: "Private and Personalized Frequency Estimation in a Federated Setting"
_NeurIPS.cc/2024/Conference — NeurIPS 2024 poster_

### Official Review · Reviewer_a5Ce · 2024-07-10

**Soundness:** 3
**Presentation:** 2
**Contribution:** 3
**Rating:** 5
**Confidence:** 3

**Summary:**

This paper studies the problem of personalized frequency estimation in federated learning. The authors propose both private and non-private algorithms based on clustering and Good-Turing estimators and demonstrate promising empirical results on real-world datasets.

**Strengths:**

1. The problem of data heterogeneity is very important in federated learning.
2. The proposed algorithms are practical and can be applied to large-scale datasets.
3. Empirical results demonstrate improvement over FedAvg and MAML baselines.

**Weaknesses:**

1. The cluster assumption requires choosing the cluster number $K$.
2. KL divergence may not work well for sparse distributions, and this is possibly why the authors need to assume $P_c[v]$ is constant for all $v$ in Theorem 5.1. The authors motivate this work by estimating the frequency of words, but they are usually very concentrated over a very small fraction of words.
3. In Theorem 5.1, the minimum KL divergence is at least order $d$ which seems restrictive at first glance, especially considering the number of words could be of order $10^4-10^5$. Also, the $k^2$ in line 266 seems to be $K^2$.
4. The $\log(1/\delta)$ in the privacy guarantee in Theorem 5.2 can be pretty large when $\delta$ is very small. Since normally we require $\delta\sim o(1/n)$ where $n$ is the number of users, this introduces a $\log n$ factor to the privacy guarantee.
5. In the experiments, the authors only show results for $\varepsilon=15$.
6. Relevant papers to cite and discuss:

[1] Ozkara, Kaan, et al. "A statistical framework for personalized federated learning and estimation: Theory, algorithms, and privacy." International Conference on Learning Representations (ICLR), 2023. OpenReview. net, 2023.

[2] Liu Y, Suresh A T, Yu F X X, et al. Learning discrete distributions: user vs item-level privacy[J]. Advances in Neural Information Processing Systems, 2020, 33: 20965-20976.

[3] Liu Y, Suresh A T, Zhu W, et al. Algorithms for bounding contribution for histogram estimation under user-level privacy[C]//International Conference on Machine Learning. PMLR, 2023: 21969-21996.

[4] Huang, Ziyue, Yuting Liang, and Ke Yi. "Instance-optimal mean estimation under differential privacy." Advances in Neural Information Processing Systems 34 (2021): 25993-26004.

=====Update=====

Increased my score to 5 after the rebuttal

**Questions:**

1. How does the choice of $K$ affect the performance? Do the authors have a suggestion on how to choose $K$ in practice?
2. In line 256 ``Without any assumptions on the relationship between the user data distributions the best one can do is to locally estimate each user’s distribution'' I believe this should be true, but could the authors provide a brief justification for this claim?
3. Why did the authors not consider using privacy accounting methods such as RDP accountant which may give much tighter privacy bounds than the composition theorem of approximate DP?

**Limitations:**

The authors addressed the limitations and negative impact of their work.

---

> ### Author Rebuttal · Authors · 2024-08-07
>
> Thank you for the feedback. To address concerns, we add new experiments with different values of $\varepsilon$, number of local points $m$, and plots showing the sparsity of the estimated cluster centers in practice. We also point to our ablations on $K$ in our paper, and clarify concerns with Thm. 5.1, 5.2. We thank you for the relevant citations, and we will use the 1 extra page in the final version to incorporate the new experiments, clarifications, and citations. **Please let us know if your concerns are addressed, and if so, we would be grateful if you might raise your score.**
>
> >> **Different values of $\varepsilon$ and local data points $m$**
>
> We show performance evaluations under varying values of $m$, and fixed privacy budget of $\varepsilon=15$ (Fig 3 in PDF), and varying $\varepsilon$ for the values of $m$ in our paper (Fig 2 in PDF). We note that while the performance of each method drops with reducing $\epsilon$ and $m$, our algorithm has better privacy-utility tradeoffs and is more sample efficient. We also note that these privacy parameters are for a relatively small dataset (10k to 50k users). If the population size was, say, 10x larger, the privacy parameters would be 10x smaller for the same kind of results. In practical FL settings, we would often expect the population sizes to be much larger and thus the epsilons to be proportionately smaller.
>
>
> >> **Choice of number of clusters $K$**
>
> In Fig. 3a (Sec 6) and Fig. 5 (App B) we show how the choice of $K$ (number of clusters) affects the final performance of our algorithm. As expected, the performance improves as $K$ roughly matches the natural heterogeneity/clusters present in the data distribution. In practice, we can use hold out to validate $K$. See our global response for more discussion on this.
>
> >> **Privacy accounting via RDP**
>
> We do not use composition results for approximate DP. All our composition results rely on zero concentrated DP (zCDP) (Bun and Steinke [16]) for privacy accounting of user-level DP. We then convert the zCDP guarantees to approximate DP in Thm. 5.2. Note that zCDP is a stricter notion of privacy than RDP since it requires Renyi divergence of multiple orders to be simultaneously bounded. Further, it enjoys the same benefits as RDP for composition, since zCDP’s sequential composition is also asymptotically tight for repeated applications of the Gaussian mechanism [16].
>
> >> **[New Experiment] Sparsity of $P_c[v]$ in practice**
>
> In practice, the estimated cluster centers are sparse, as we see on Reddit and StackOverflow, where about 10% of the tokens carry 90% of the probability mass (Fig 4 in 1-page PDF). The assumption we make in Thm. 5.1 on P_c[v] being lower bounded by a constant is needed solely for the theoretical convergence guarantees for clustering. This is needed in theory because KL divergence does not obey triangle inequality, but $l_2^2$ distance does, and for KL between estimated to be roughly tracked by $l_2^2$, we need $P_c[v] = \Omega(1)$.
>
> >> **Dependence on $d$ in the KL separation condition in Thm. 5.1**
>
> There is a typo in the condition on the minimum cluster separation $\Delta$. The correct condition is $\Delta = \Omega(k^2 + \frac{k^3 d}{n})$, as derived in the proof of Thm. 5.1 in  App. D.1 (below L831). We apologize for this typo and will correct it in the final version. Note that the number of clients $n$ appears in the separation condition on $\Delta$, and the convergence is determined by the condition number. Thus, the condition on KL divergence only scales as $poly(k)$ when $n=\Theta(d)$. For the datasets we use, the number of users is sufficiently large ($d/n < 2$ for all).
>
> >> **$\log(1/\delta)$ term in Thm. 5.2**
>
> Yes, for the privacy guarantee to be meaningful $\delta$ needs to be $o(1/n)$. But, note that the guarantee in Thm. 5.2 is $\sqrt{\log(1/\delta)}$, which for the datasets of our size (around $10^5$), translates to $\approx \sqrt{5}$.
>
> To elaborate more on the term, the result in Thm. 5.2 is obtained by first composing the privacy mechanisms in each clustering iteration with zero-concentrated DP (zCDP) and then converting the zCDP guarantees into approx. DP using Lemma 3.6 in Bun & Steinke [16]. The $\log (1/\delta)$ term is an artifact of this conversion. On the other hand, if we were to instead compute approx. DP guarantees (instead of zCDP) at each clustering iteration and then use the advanced composition theorem to compose them, we would still end with a $\sqrt{\log(1/\delta)}$ term from the composition. So, this term is unavoidable either way.
>
>
> >> **L256 Without any assumptions on the relationship … the best one can do is locally estimate**
>
> Consider the following worst-case setting where there is no mutual information between the training datapoints sampled by any pair of users. Let each user’s true token distribution $Q_u$ be supported over subsets of tokens that are disjoint for any pair of users. Further, for any user $u$ and token $v$ with $Q_u[v]>0$, the value of $Q_u[v]>0$ is independent of the token distribution of other users. In such a case, no estimator can improve the average estimation error through collaboration/sharing of data. The optimal estimator is purely local. We will add this example to the paper.
>
> >> **Missing citations**
>
> Thanks for pointing out these references, we will cite & discuss them. We believe our contributions go beyond these: for e.g., while [2, 3, 4] look at private histogram estimation, they concern with algorithms that estimate a single global quantity (e.g., mean histogram) whereas ours deals with the nuances of privatizing personalization algorithms that estimate a separate distribution for each user, and learn multiple global centers. Further, [1, 4] conduct their analysis in the $l_2$ metric, whereas we look into the more appropriate KL distance for distribution estimation. KL is harder to analyze and develop algorithms for, given that it does not even obey the triangle inequality.

---

> > ### Author Response · Authors · 2024-08-13
> > **Request for discussion**
> >
> > Dear reviewer,
> >
> > Apologies for bothering you! Since we only have one day left in the discussion period, we would be grateful and would sincerely appreciate if you could respond to our rebuttal, leaving us enough time to address any remaining questions.
> >
> > Thanks,
> > Authors

---

### Official Review · Reviewer_DFyV · 2024-07-14

**Soundness:** 3
**Presentation:** 3
**Contribution:** 2
**Rating:** 6
**Confidence:** 3

**Summary:**

This work introduces a private and personalized histogram estimation approach based on Good-Turing estimates, when the clients are clustered. Theoretically, they provide performance guarantees for different settings when cluster center is known and cluster assignments are known. In practice, they note that neither cluster centers nor assignments are known, hence inspired by theoretical results from the simpler settings they provide practical algorithms. Later they also introduce differential private version of the proposed methods whose performance can increased further with a particular initialization method. For experiments they consider three tasks where the proposed method outperforms the competition.

**Strengths:**

-Private and personalized histogram estimation is an important problem.

-The paper is easy to understand, presentation and justification of the methods are well organized.

-The experiments are conducted on multiple datasets, the method consistently outperforms others.

**Weaknesses:**

- Lack of comparison to local only training and other personalized estimation methods.

- The main results are rather in simpler settings, the analysis mostly builds on top of previous work; hence, theoretical contribution might be limited (compared to methodological).

- From the current experiments, performance dependence on variables such as number of clusters, clients, data samples per clients, heterogeneity etc. is not clear. I would suggest doing more ablations studies possibly using e.g. synthetic datasets to understand these.

**Questions:**

See weaknesses.

**Limitations:**

Yes

---

> ### Author Rebuttal · Authors · 2024-08-07
>
> Thank you for the feedback and a positive assessment of our work. To address your concerns, we present new ablations of our algorithms under different values of per-client training data points $m$, local only training baselines, highlight existing ablations in our paper on the number of clusters, and address the point of analyzing a simpler setting. We will add the new results and discussion on the above points in the extra page we get for the final version. **Please let us know if your concerns are addressed, and if so, we would be grateful if you might raise your score.**
>
>
> >> **[New experiment] Lack of comparison to local only training and other personalized estimation methods.**
>
> In Fig 5 (1 page PDF), we find that the local only training performance where the clients never collaborate is worse than our approach (for a privacy budget of $\epsilon=15$), even as we increase the number of local data points $m$. Further, comparing this result to Fig 2 (1 page PDF), we find this trend to hold even under a very low privacy budget of $\varepsilon=2$.
>
>
> >> **The main results are rather in simpler settings, the analysis mostly builds on top of previous work.**
>
> While our analysis technique adapts results on clustering and Good-Turing estimation to our setting, its main goal is to serve as a guide that formally validates our algorithmic choices, for e.g. the local finetuning algorithm (in Eq. 2), or using the average of local Good-Turing estimates for estimating the cluster center, as opposed to the typical average of empirical estimates that is used in the K-means algorithm (see our comparison in Thm. 4.3). In the FL and privacy literature, it is also common for algorithms to be analyzed under such models of data heterogeneity (e.g., Cummings et al. [22], Cheng et al. [20]) that mimic real world distributions.
>
> >> **[New experiment] From the current experiments, performance dependence on variables such as number of clusters, clients, data samples per clients, heterogeneity etc. is not clear.**
>
> **Number of clusters**: In Fig. 3a (Sec 6) and Fig. 5 (App B) we show how the choice of $K$ (number of clusters) affects the final performance of our algorithm. As expected, the performance improves as $K$ roughly matches the natural heterogeneity/clusters present in the data distribution. In practice, we can cross-validate to find $K$ and use $K=10$ (close to the optimal choice for all datasets), for all our runs.
>
> **Number of clients and privacy budget**:In the attached PDF, we show performance evaluations under varying values of $m$ and fixed privacy budget of $\epsilon=15$ (Fig 3), and varying $\epsilon$ for the values of $m$ in our paper (Fig 2). We note that while the performance of each method drops with reducing $\epsilon$ and $m$, our algorithm has better privacy-utility tradeoffs and is more sample efficient
>
> **Heterogeneity**: Given that the natural datasets we consider already have varying levels of heterogeneity (see discussion in global response on a different choice of $K$ being optimal on each dataset) and allow us to investigate the key phenomenons, we see no compelling reason to empirically study synthetic data. We are also not aware of a sufficiently faithful model of data for our problem (and even for our own data model there are many parameters, for which it is unclear how to choose synthetic values). At the same time we welcome and will certainly consider specific suggestions about synthetic data experiments that would make our work more informative.

---

> > ### Comment · Reviewer_DFyV · 2024-08-10
> >
> > Thanks for the rebuttal, I acknowledge reading the rebuttal and will keep my score.

---

> > > ### Author Response · Authors · 2024-08-13
> > >
> > > Dear reviewer,
> > >
> > > Thank you for your acknowledgement and positive assessment of our work. We will definitely add the new ablations on data samples per client, different privacy budgets and related discussion to the main paper.
> > >
> > > Since there is still one more day, we are also wondering if there would be some other discussion or evidence that we can provide in this period to help improve your evaluation of our paper further. Please let us know. Thanks a lot!
> > >
> > > Thanks,
> > >
> > > Authors

---

### Official Review · Reviewer_zabr · 2024-07-26

**Soundness:** 3
**Presentation:** 1
**Contribution:** 2
**Rating:** 5
**Confidence:** 3

**Summary:**

The paper presents both non-private and private per-user histogram estimation approaches that can be applied to problems such as next word predictions. The approach relies on iteratively clustering among different users to find similar user subgroups and then fine-tuning within each user to get user-specific estimations. The privacy guarantee is achieved by privately finding clusters using all users data and is quantified by the notion of joint-differential privacy. The paper theoretically analyzes the sub-optimality of their algorithm and empirically evaluate the performance by comparing to several baselines.

**Strengths:**

- The paper empirically demonstrates that the proposed method have better performance than the baselines.
- The authors also conducted thorough ablation experiments to explain the importance of various sub-parts of the algorithm.

**Weaknesses:**

- The largest concern to me is the presentation clarity of the paper.
    - In section 4, the authors seem to present many types of assumptions regarding possible user data distributions, but the sub-titles are somewhat confusing as it tries to explain model, learning, different clustering cases, histogram estimators and also theorems. It would be much clearer if more relevant parts are put together.
    - It would be easier to understand why these assumptions are stated if in the experiment section the authors could include whether certain dataset fall into which category of the assumed data structure/distributions, and in the algorithm section how the algorithms are adapt to these cases of data structure/distributions.
    - It was also unclear to me regarding the final pipeline (the 'putting it all together' section). For example, to my understanding Algorithm 3 seems to be the 'main' algorithm where the cluster centers are determined, then to get user-specific estimations (the $\hat{Q}_u$s) do users fine-tune in their own ways or follow Algorithm 1 or something else?
- The novelty seems not very strong as the clustering part seems to be the major contribution but the papers apply standard K-means like clustering algorithms except changes to KL as the target is to minimize some KL as in Equation 5.

**Questions:**

- In section 4 one of the assumption is 'the user histograms are well concentrated along each token’s marginal', what does 'each token’s marginal' means and can it be described in mathematical language in terms of 'well concentrated'?
- Since the authors defined their own DP relaxation definition, it will be helpful if the authors could clarify:
    - Is there a conversion between scale of $\epsilon$ to the pure or approximate DP case? The $\epsilon=15$ value reported in the evaluation section seems to be a quite large number for meaningful privacy protection, how should it be interpreted under the $\rho$-zCJDP notion?
    - I did not completely understand the $\rho$ parameter in Definition 3.2. Is it another relaxation defined if it's defined as the divergence term $D_{\alpha} < \rho \alpha$?

**Limitations:**

see above

---

> ### Author Rebuttal · Authors · 2024-08-07
>
> Thank you for the feedback! To address concerns we provide an algorithm’s box for the end-to-end algorithm, clarify that assumptions in Sec 4 are only for theoretical motivation, and results in Sec 6 are on real datasets where they needn't hold. We clarify our DP definition is not relaxed at all, in fact it is more stringent than RDP and the final guarantees in Thm. 5.2 are for the approx DP definition. We will use the 1 extra page in the final to incorporate them. **Please let us know if your concerns are addressed, and if so, we would be grateful if you might raise your score.**
>
> >> **Final alg in Sec 5**
>
> In the 1 page PDF (Fig 1), we show an algorithm box for the end-to-end algorithm that uses Alg 5 for private initialization of cluster centers, and Alg 3 for clustering, where the re-centering step uses Alg 4. Each user finetunes their corresponding cluster center using Eq. 2.
>
> >> **Assumptions in stylized model and concentration of user histograms**
>
> - In Sec 4, to guide algorithm design we introduce a model where the user distributions are distributed as a mixture of Dirichlets. This model has two latent variables: cluster memberships and cluster centers. For simplicity, we analyze estimators under increasing levels of knowledge in this model: cluster center known -> cluster center unknown but cluster membership known -> both cluster center and membership unknown.
> - The simpler analysis in a clean model leads us to three key findings (end of Sec 4) using which we propose the clustering algorithm (Alg. 3), with the recentering step using Good-Turing estimates (Eq. 4), and finetuning given by Eq. 2.
> - In Sec 6, we evaluate our proposed algorithm in the most general setting, where cluster centers/memberships are unknown. Here, we also do not assume anything about the distribution of $Q_u$’s.
> - Regarding the **well-concentrated** assumption: We only need that w.h.p., for all users $u$ in cluster $c$, and any token $v$, $|P_c[v] - Q_u[v]| = O(1)$. This is naturally satisfied by our Dirichlet assumption since the $\alpha$ parameter of Dirichlet allows us to analyze our setting with varying degrees of cluster-level concentration where $|P_c[v] - Q_u[v]| = O(1/\sqrt{\alpha})$.
>
>
> >> **Novelty**
>
> The novelty of our Algorithm lies beyond the proposed K-means style clustering method. Following is a set of novel contributions we make, where the final four claims are verified empirically in Sec 6 and defined, analyzed formally in Sec 5.:
> - Introduce the personalized and private frequency estimation problem in KL divergence.
> - Contrary to popular beliefs (Cheng et al. [20], Wu et al. [76]) in the FL community, we show that there exists practical FL problems where finetuning a single global model (FedAvg+FT) is not sufficient for personalization.
> - Specifically to improve clustering error in terms of KL divergence (as opposed to squared loss in k-means) we motivate and propose a Good-Turing based center estimator (Sec 4).
> - Propose a private version of the clustering algorithm with approximate differential privacy guarantees (Thm. 5.2).
> - Propose a novel data-dependent and yet private initialization for clustering (Alg. 5), which is crucial to improve clustering performance (Fig. 3b). Prior works on private-clustering (e.g., Chang et al.: Locally private k-means) do not analyze expectation maximization (EM) algorithms given their dependence on initialization which is harder to privatize (most initializers for EM directly output a set of points from the dataset (Arthur et al. [5])).
> - Propose a practical 2-stage algorithm for the size-heterogeneous setting (Section 5). Naive algorithms in this setting typically hurt the privacy-utility trade off since hiding the participation of data-rich users now becomes very hard when other users have less data. We show that our proposed algorithm is able to improve estimation error from including both data-poor and data-rich users in the collaboration, without hurting the privacy utility trade off (Fig 4).
>
>
> >> **Our final guarantees are in terms of approx DP and Def 3.2 is stricter than RDP**
>
> Our final guarantees in Thm 5.2 are in terms of user-level $(\varepsilon, \delta)$-DP which is equivalent to JDP in our setting. We only use the $\rho$-cJDP (stricter than RDP) in Def 3.2 for easier and tighter composition (than approx. DP), and provide final guarantees in terms of user-level approx. DP.
>
> **In billboard model, JDP=user-level DP**: Note that our setting is akin to the billboard model (Hsu et al. [41]) where any $(\varepsilon, \delta)$ user-level DP (Dwork [25]) algorithm is also $(\varepsilon, \delta)$-JDP. Similarly, if the algorithm satisfies user-level $\rho$-zCDP (Bun et al. [16]) then, it is also ρ-zCJDP (Def 3.2).
>
> **Final guarantees are for user-level DP**: We use zCDP for composition at the user-level, and then use Lemma 3.6 from [16] to convert zCDP to approx. DP in Thm. 5.2.
>
> **Why use zCDP?**: Yes, zDP bounds the $\alpha$ Renyi divergence between the mechanism outputs on neighboring datasets by $\rho\alpha$. Thus, zCDP is a stricter notion than Renyi DP (RDP) since zCDP requires Renyi divergence of multiple orders to be simultaneously bounded. Further, zCDP enjoys the same benefits as RDP for composition, since zCDP’s sequential composition is also asymptotically tight for repeated applications of the Gaussian mechanism [16].
>
> >> **[New Experiment] Evaluating performance at different privacy budgets**
>
> In Fig 2 (1 page PDF) we evaluate our algorithm and other baselines under different privacy budgets. Each algorithm needs to satisfy ($\varepsilon, \delta$)-JDP (which is ($\varepsilon, \delta$) user-level DP). On the x-axis we vary $\varepsilon$ and fix the same $\delta$ reported in the paper. We find that our approach continues to outperform baselines. While the clustering baseline IFCA and MAML suffer from severe performance degradation under low privacy budgets, our approach observes a much better privacy-utility trade off.

---

> > ### Comment · Reviewer_zabr · 2024-08-12
> > **Reply to Author Rebuttal**
> >
> > Thanks for the explanations especially on the DP guarantees, it makes sense to me.
> > - Regarding clarity on Section 4: I think ``cluster center known -> cluster center unknown but cluster membership known -> both cluster center and membership unknown'' makes sense, but the current section 4 lacks such structure which makes it difficult to understand which result/findings is linked to which setting. Since the authors mention Section 4 is for theoretical motivation and Section 6 tests the most general setting only, I would expect the authors to highlight and elaborate on the key findings that lead to the design of the actual algorithm.
> > - Regarding novelty and experiments: I appreciate the new experiments added, but my main concern still remains which the paper proposes an algorithm that targets to minimize the KL term between users' histogram and the estimated cluster center histogram distributions, which is also the metric to evaluate in experiments, but the baselines are not designed for the same problem. For example, FedAvg, MAML and IFCA are federated/distributed deep learning frameworks that train better models in federated setting on regression/classification tasks, are gradient-based, and are not particularly privacy-preserving designed. I would expect authors to compare to more similar approaches in private frequency estimation such as [1, 2].
> >
> > For the above reasons I would keep my score.
> >
> > [1] Private Federated Frequency Estimation: Adapting to the Hardness of the Instance. Wu et al. NIPS 2023. https://arxiv.org/abs/2306.09396.
> > [2] The communication cost of security and privacy in federated frequency estimation. Chen et al. AISTATS 2023. https://arxiv.org/abs/2211.10041.

---

> > > ### Author Response · Authors · 2024-08-12
> > > **Choice of Baselines and New experiments with private frequency estimation baselines**
> > >
> > > Thank you for your comments and suggestions for private global frequency estimation baselines. We add new experiments comparing our method with suggested baselines [1, 2], and find that for all ranges of the privacy budget our approach outperforms [1, 2], highlighting the need for learning clusters and cluster-level frequency counts (as opposed to global counts) to reduce the estimation error in KL, per-client. We also clarify that the baselines (FedAvg/MAML/IFCA) we used in our paper were originally proposed for more general settings, and we adapt each of these baselines in a fair manner for a meaningful comparison. We will add these experiments and discussion to the final version. **Please let us know if this addresses all outstanding concerns, and if so, we would be grateful if you might raise your score.**
> > >
> > > ___
> > >
> > > **Adapting FedAvg/MAML/IFCA, which are proposed for convex optimization (superset of our setting which is to minimize KL distance):** We will make this more clear, but please note that all three baselines have been proposed and analyzed as general algorithms to solve convex objectives (our metric of KL divergence is also convex) in the federated setting. We apologize if this is not clear, but for a fair comparison, we adapt the baselines specifically to our setting in the following way:
> > >
> > > - FedAvg and MAML both optimize the KL distance (as opposed to squared loss) to learn a single distribution that is close to the empirical distributions of each user in KL distance (Eq. 3 in our paper). Note that we do not use gradient-based approaches for FedAvg, and instead use the closed-form optimal (in Eq. 3, proof in Appendix C).
> > >
> > > - IFCA (Ghosh et al. [35]) is one of the most popular clustering baselines in FL, that was proposed for convex losses, and analyzed for k-means (squared loss), which we adapt to our setting by: 1) using the KL distance to the current cluster centers for determining the cluster affiliation of each user in a clustering round; and 2) estimating the new cluster centers by optimizing the KL objective, which is same as what we did for FedAvg/MAML but now only for clients in the cluster (Eq. 5 in our paper).
> > >
> > > - Estimates from all three baselines are personalized (adapted) for each user using the same method as ours (Eq. 2), which is not gradient-based. Additionally, we also try other popular approaches like RTFA, gradient-based (Fig. 3d) and find our non-gradient based approach to work best for all three baselines. The local adaptation step for MAML also uses the same non-gradient based method.
> > >
> > > ___
> > >
> > > **Comparison with private frequency estimation approaches [1, 2]**: Thank you for suggesting these baselines. Wu et al. [1] is based on the same algorithm (Count Sketch+Secure Aggregation) as Chen et al. [2], and mainly proposes a two-stage approach that adapts to the hardness of the problem instance (heterogeneity of the frequency counts). Both papers [1, 2] differ from our setting in the following ways:
> > >
> > > 1) They estimate a single global frequency count (FedAvg estimate in our case) in $l_1$ or $l_\infty$ norms, but our problem setting requires estimating personalized distributions for each user (and that too in KL divergence which is not a proper distance metric).
> > >
> > > 2) They mainly use CountSketch to encode the local frequencies to reduce the per-round communication cost.
> > >
> > > _[1, 2] are concerned with communication cost:_  In the best case, when there is no communication constraint, their approach resorts to only using secure aggregation (which uses Gaussian mechanism in [1] and Poisson Binomial in [2]), and is thus same as ours which uses Gaussian Mechanism (Alg. 4). We are not concerned with reducing the communication cost, but if we were, it is straightforward to use the CountSketch algorithm at each round of clustering. Further, note that while CountSketch reduces the number of bits communicated by each client, the error suffered on each item's frequency (due to privacy noise) would not change at all. **So CountSketch can only improve communication complexity over our method and not privacy-utility tradeoff**.
> > >
> > > _Similar to [2], our approach also adapts to the range/hardness:_ In algorithm 4, for private center estimation for the global center (in FedAvg) or cluster-level center (in Alg. 3), we use the common adaptive clipping technique which first uses some privacy budget to estimate the range of the true center to a small confidence interval. Then, the user-level estimates are clipped to this small confidence interval before running secure aggregation with the rest of the privacy budget. Thus, **the adaptive clipping we use is similar to the two-stage approach in [2], in the sense that both adapt to the hardness of the underlying instance.** We find this interesting and will add this comparison to the main paper.
> > > ___
> > > **PLEASE SEE OUR CONTINUATION BELOW FOR OUR EXPERIMENTAL RESULTS**

---

> > > > ### Author Response · Authors · 2024-08-12
> > > > **New experiments with private frequency estimation baselines**
> > > >
> > > > **[NEW EXPERIMENTS] Fair comparison with suggested private frequency estimation approaches [1, 2]:**
> > > >
> > > > We compare our approach with the suggested baselines [1, 2], after we do the following:
> > > >
> > > > 1) tune the sketch length hyper-parameter $L$ (over 2, 4, 8, 16, 32) and set $L=8$ and tune sketch width $W$ (over 10,  20, 50, 100, 200, 500, 1000) and set $W=500$.
> > > >
> > > > 2) adapt the estimated global frequency (distribution) for each user using our finetuning algorithm (Eq. 2), that we also use to adapt cluster centers/FedAvg estimate for each user separately.
> > > >
> > > > We use the same setup (number of clients and local data points per client) as our main experiments in Figure 2, except for the privacy budget $\varepsilon$, which we additionally vary from 4 to 16.
> > > >
> > > > ___
> > > >
> > > > | StackOverflow Test NLL Loss          |$\varepsilon=4$ | $\varepsilon=8$ | $\varepsilon=16$ |
> > > > |--------------------------------------|-------|-------|--------|
> > > > | CountSketch + Secure Aggregation [1] | 9.935 | 9.399 | 9.017  |
> > > > | 2-stg CountSketch + Secure Aggregation [2] | 9.783 | 9.132 | 8.755  |
> > > > | Ours                                 | **9.008** | **8.544** | **8.107**  |
> > > >
> > > > ___
> > > >
> > > > | Reddit Test NLL Loss                 | $\varepsilon=4$ | $\varepsilon=8$ | $\varepsilon=16$ |
> > > > |--------------------------------------|-------|-------|--------|
> > > > | CountSketch + Secure Aggregation [1] | 7.618 | 7.012 | 6.806  |
> > > > | 2-stg CountSketch + Secure Aggregation [2] | 7.427 | 6.914 | 6.785  |
> > > > | Ours                                 | **7.013** | **6.589** | **6.312**  |

---

> > > > > ### Comment · Reviewer_zabr · 2024-08-13
> > > > > **Reply to Author Rebuttal**
> > > > >
> > > > > I thank the authors for their quick response and for adding new experiments. I have raised my score. However, given the discussion above I think the paper needs major revision to clarify the theoretical motivations for designing algorithms in Section 4, implementation details to compare baselines that are not closely targeting the same problem, and potentially add a clearer related work section and compare to work that are more closely related to the field of private histogram release. I think that is the best score I am confident to raise given the current version.

---

> > > > > > ### Author Response · Authors · 2024-08-13
> > > > > >
> > > > > > Thank you for raising the score. In the final version, using the extra page, we are certainly committed to provide more clarity on our theoretical motivation for algorithm design in Section 4. We will also add results obtained with the new baselines in the discussion above, further clarify the implementation details of existing baselines in the paper and add missing citations to the related work section. Thank you again for your detailed feedback and comments.

---

### Official Review · Reviewer_ymNE · 2024-08-02

**Soundness:** 3
**Presentation:** 2
**Contribution:** 3
**Rating:** 7
**Confidence:** 3

**Summary:**

The paper proposes a federated learning approach for frequency histogram estimation in a distributed setting. The proposed approach relies on first clustering users that have similar subpopulation distributions before performing the estimation in a privacy-preserving manner. Finally, the performance of the approach is also validated on three real-world datasets.

**Strengths:**

The paper is well-written and the setting considered in the paper is clearly introduced. The challenges related to frequency estimation in the federated learning setting as well as the related work are also reviewed in a thorough manner.

One of the novelty of the proposed approach is the idea of clustering users first before performing the frequency estimation. The authors that the proposed approach offers strong theoretical guarantees and does not suffer from the same limitations as other methods from the state-of-the-art, such as the sensitivity to small datasets.

Based on the proposed approach, the authors have developed several variants of the personalized frequency estimation histograms, including for the private and no-private settings as well as for the size-heterogenous one. The private version of the algorithm combines several clever privacy-preserving subroutines in a clever manner.

The experiments conducted are quite extensive as they have been done on three datasets and have compared the approach to three different baselines.

**Weaknesses:**

The paper focuses on next word prediction, which is a quite focused application. Instead, it would be more interesting to try to broaden the applicability of the approach, in particular by testing it on other type of datasets. The possibility to extend the framework developed beyond the KL-divergence should also be discussed.

The neighbouring notion considered for differential privacy seems quite strong in terms of the privacy guarantees provided but the authors failed to motivate why this is right notion to consider in the federated learning in comparison to record level differential privacy. The necessity of the Dirichlet assumption as well as the concentration of the users’ histograms should also be justified.

A discussion on the method that should be used to estimate the number of clusters K is currently missing in the paper. This is a crucial point as this parameter is likely to have an important impact on the success of the approach.

In terms of the structure of the paper, it ends quite abruptly lacks a conclusion summarizing the key findings of the paper and also discussing possible future works.

A few typos :
-« Baysian prior » -> « Bayesian prior »
-« privacy preserving analysis » -> « privacy-preserving analysis »
-« estimatation in KL divergence » -> « estimation in KL divergence »
-« For competetive estimator » -> « For competitive estimator »
-« LLoyd’s K-means » -> « Llloyd’s K-means »
-« differentially private » -> « differentially-private »
-« Privacy preserving noisy sum » -«  « Privacy-preserving noisy sum »

**Questions:**

What is the relationship of the privacy parameter rho compared to standard parameters epsilon and delta in differential privacy?

Please see the main points raised in the weaknesses section.

**Limitations:**

The authors have a specific section discussing the limitations of their work.

---

> ### Author Rebuttal · Authors · 2024-08-07
>
> Thank you for the feedback and a positive assessment of our work. To address concerns, we refer to plots in the paper that ablate $K$, clarify guarantees in Thm. 5.2 are in terms of the typical $(\varepsilon, \delta)$ approx. DP, and discuss the choice of the stricter user-level DP definition. We also explain the motivation to study next-word prediction, and the choice of the stylized model: mixture of Dirichlets. Thank you for pointing us to typos, we will correct them, add discussion on the above points and a conclusion section in the extra page we get for the final. **Please let us know if your concerns are addressed, and if so, we would be grateful if you might raise your score.**
>
> >> **Choosing the number of clusters K**
>
> In Fig 3a, 5 in our paper, we plot the test NLL when varying the number of clusters $K$ parameter for our clustering Alg 3. **Please see the global response for more discussion on this**.
>
> >> **Converting cJDP to user-level DP**
>
> Our final guarantees in Thm 5.2 are in terms of user-level $(\varepsilon, \delta)$-DP which is equivalent to JDP in our setting. We only use the $\rho$-cJDP (stricter than RDP) in Def 3.2 for easier and tighter composition (than approx. DP), and convert them to user-level approx. DP finally.
>
> Our setting is akin to the billboard model (Hsu et al. [41]) where any $(\varepsilon, \delta)$ user-level DP (Dwork [25]) algorithm is also $(\varepsilon, \delta)$-JDP. Similarly, if the algorithm satisfies user-level $\rho$-zCDP (Bun and Steinke [16]) then, it is also ρ-zCJDP (Def 3.2). Thus, we first use zero-concentrated DP (zCDP) for privacy accounting and composition, and then use Lemma 3.6 from [16] to convert the zCDP guarantees to approx. DP in Thm. 5.2.
>
> >> **User vs. record level DP**
>
> We defined our problem in Sec. 3 with the goal of achieving personalized predictions for each user’s frequency counts, while satisfying user-level privacy. As you correctly point out, this is indeed stricter than item-level privacy, but it is also a common choice for privacy guarantees in cross-user federated learning (e.g., Cummings et al. [22], Levy et al. [52], Wang et al. [71]). In cross-user FL, e.g. learning to predict the next word on users’ mobile devices, it is natural to protect the participation of every user; as a user may type many different words that are correlated, hiding a single word may not be sufficient privacy protection. Indeed if we had the weaker item level DP protection with $\varepsilon = 2$ (say), the overall user-level privacy loss for a user that contributed $m=200$ words would theoretically be $\varepsilon \cdot m = 400$. Even if 10 of the words are correlated with sensitive information (e.g. related to their medical diagnosis), the $\varepsilon$ parameter increases by that factor. Our user-level guarantee ensures that even if the user were to change all 400 words they contributed, the models will not be significantly different.
>
> >> **Using the mixture of Dirichlets model for our analysis in Sec 4**
>
> In Sec. 4, we introduce the mixture of Dirichlets model with the goal of guiding algorithm design when the underlying population is heterogeneous, yet is structured. While each user has a different token distribution it is likely that there exists clusters of ``similar’’ users. This model motivates a setup where adapting a single global model for each user would be less effective than identifying clusters, learning a single model for the cluster and then adapting the cluster-level model for the user. We also observe this empirically in our experiments in Sec 6, where FedAvg+FT performs worse than our approach.
>
> Note that the specific choice of Dirichlet for analysis is only so that we can easily derive the optimal Bayes estimator (Thm 4.2), which gives us our finetuning algorithm in Eq 2. For all other theoretical results in Sec. 4, 5 we only need each user’s true distribution to be concentrated along the cluster center, i.e., w.h.p., for all users $u$ in cluster $c$, and any token $v$, $|P_c[v] - Q_u[v]| = O(1)$. The $\alpha$ parameter of Dirichlet further allows us to analyze our setting with varying degrees of cluster-level concentration, since under the Dirichlet assumption $|P_c[v] - Q_u[v]| = O(1/\sqrt{\alpha})$.
>
> >> **Why focus only on the next-word prediction problem in the FL setting? Extending to other datasets and losses.**
>
> Large language models are next-token prediction models that can be used to solve tasks in natural language like math reasoning, coding etc. In FL, the performance of these next-token prediction models can be improved fairly across users when they are personalized (Hard et al. [38], Salemi et al. [66]). At the same time, since each user has few data points for purely local learning, global collaboration is needed to bias the local learning algorithm. Thus, we study frequency estimation (estimating the marginal distribution) as a first step towards private and personalized language models. We highlight how measuring the error in KL divergence and reducing privacy-utility trade offs requires algorithmic innovations (e.g., clustering with Good-Turing, data-dependent private initialization) that are specific to our problem and much required.
>
> We provide an extended analysis on three real-world datasets with varying levels of statistical heterogeneity, each of which are a popular choice for analyzing next-token predictors in FL (Wu et al. [76]). Please let us know if there is a specific dataset that is characteristically different from these, and we would be happy to include experiments on it.
>
> As we mention in Sec 1, we focus on error measured in KL as is common in language modeling (Hoffmann et al. [40]), and is equivalent to minimizing the negative log-likelihood of the sample. Distribution estimation in KL distance is also studied more generally (Drukh et al. [24], Acharya et al. [2]). We believe that extending KL to other losses is an interesting direction of future work, as opposed to a weakness.

---

> > ### Author Response · Authors · 2024-08-13
> > **Request for discussion**
> >
> > Dear reviewer,
> >
> > Apologies for bothering you! Since we only have one day left in the discussion period, we would be grateful and would sincerely appreciate if you could respond to our rebuttal, leaving us enough time to address any remaining questions.
> >
> > Thanks,
> > Authors

---

> > > ### Comment · Reviewer_ymNE · 2024-08-14
> > >
> > > Thanks for your rebuttal. I appreciate the clarifications that have been provided and as well as the addition experiments and I will increase slightly my score.

---

> > > > ### Author Response · Authors · 2024-08-14
> > > >
> > > > Thank you for increasing the score. We are committed to include the above discussion and clarifications in the final version. Thank you again for your thoughtful feedback and comments.

---

### Author Rebuttal · Authors · 2024-08-07

We thank all reviewers for their feedback and list some of the new experiments we add to the 1-page PDF which we will incorporate using the extra page in the final version. We also address a common concern on the choice of number of clusters $K$ for our Algorithm 3.

___
## **List of new experiments in 1-page PDF.**
- Figure 1: Outline of the end-to-end algorithm that first privately initializes cluster centers and then runs private clustering. Finally, the learned cluster centers are finetuned by each user locally.
- Figure 2: We evaluate our method and baselines under different privacy budgets (varying $\varepsilon$), where each algorithm needs to satisfy $(\varepsilon,10^{-10})$-JDP. We find that our approach continues to improve over baselines even at low privacy budgets and has better privacy-utility tradeoff.
- Figure 3: We evaluate our method and baselines under different values of the number of local datapoints $m$ and find that our approach is more sample efficient than baselines.
- Figure 4: We plot the CDF plot for the distribution over tokens given by the cluster centers estimated on Reddit and StackOverflow. The plots suggest that 90% probability mass is present in 10% of the tokens indicating that the underlying cluster centers are indeed sparse. But note that the same is not true for the FedAvg estimate indicating that the global center is denser. This means that the discovered underlying clusters are sparse and diverse.
- Figure 5: We evaluate our method at $\varepsilon=15, \delta=10^{-10}$ privacy budget, compared with perfectly private local only training when local dataset size is varied. Even at higher values of $m$ our algorithm is able to learn a more accurate estimate by first collaborating with users to privately learn cluster centers and then adapting them locally for each user using their local datasets only.

___
## **Choosing the number of clusters $K$ in practice.**

In Fig 3a and Fig 5 in our original submission we plot the performance of our end-to-end algorithm as we vary the choice of number of clusters $K$ used in the private initialization (Algorithm 5) and histogram clustering (Algorithm 3). We use a hold out validation set on the Reddit dataset to identify the number of clusters $K=10$ to be the optimal choice (Fig 5). We use the same choice for the other two datasets: Amazon Reviews and StackOverflow and a post hoc reveals that $K=10$ is close to optimal for those as well (Fig 3a). In fact we note that the choice of number of clusters K is not very sensitive close to the optimal. Furthermore, we also note the varying levels of heterogeneity in the different datasets we evaluate as $K=8$ and $K=16$ does better for StackOverflow and Amazon Reviews respectively. In practice, we show that one can use a hold-out validation set to more accurately judge the choice of this hyperparameter (Fig 5).

---

> ### Author Response · Authors · 2024-08-10
> **Request for discussion**
>
> Dear reviewers,
>
> Apologies for bothering you! Since we are getting close to the end of the discussion period, we would be grateful and would sincerely appreciate if you could respond to our rebuttal, leaving us enough time to address any remaining questions.
>
> Thanks,
> Authors

---

### Decision · Program_Chairs · 2024-09-25

**Decision:**

Accept (poster)

**Comment:**

The reviewers were reasonably enthusiastic about the paper. There were a couple of reviewers on the fence. We would recommend the authors to take care of the concerns raised in the reviews, in particular, w.r.t. the presentation (especially Section 4), and w.r.t. the baselines chosen for the experiments. We recommend the authors to meticulously incorporate the rebuttal discussion in a future version of the paper.